# NoTVLA: Narrative Outline of Dense Action Trajectories for Generalizable Robotic Manipulation

## Abstract

A common method for creating Vision-Language-Action (VLA) models involves fine-tuning pre-trained Vision-Language Models (VLMs) for robotic control. However, this adaptation process often leads to **catastrophic forgetting**, where the VLM's original powerful reasoning capabilities are degraded. We identify that this issue stems from a fundamental task conflict: fine-tuning on dense, continuous action trajectories is misaligned with the VLM's pre-training objectives. To tackle this, we propose the **Narrowing of Trajectory VLA (NoTVLA)** framework, which mitigates catastrophic forgetting by reframing the action generation task. Instead of dense trajectories, NoTVLA learns to predict sparse, semantically meaningful trajectory 3D points leading to keyframes. This approach aligns the fine-tuning task more closely with the VLM's inherent strengths, preserving its reasoning abilities. A key innovation of NoTVLA lies in its trajectory planning strategy, which uses temporal compression and spatial pruning for the robot end-effector's path. In multi-task evaluations, NoTVLA achieves superior performance and generalization compared to baselines like $\pi_0$, while using over an order of magnitude less compute and not necessarily need wrist-mounted camera. This design ensures that NoTVLA's operational accuracy closely approximates that of single-task expert models. Crucially, by mitigating catastrophic forgetting, it preserves the model's inherent language capabilities, enabling **zero-shot generalization** in specific scenarios, supporting unified model deployment **across multiple robot platforms**, and fostering generalization even when **perceiving tasks from novel perspectives**.

## 1 Introduction

Vision-Language-Action (VLA) models have emerged as a transformative force for embodied intelligence, heralding a new era of integrated perception, reasoning, and control (Kim et al., 2024; Brohan et al., 2023; Driess et al., 2023; Alayrac et al., 2022). By coupling large-scale multimodal reasoning engines with action experts, these systems aspire to endow robots with the capacity to operate robustly in open-ended, unstructured environments. Yet, despite rapid progress, a significant obstacle impedes reliable deployment: catastrophic forgetting (French, 1999; Kirkpatrick et al., 2017; Zenke et al., 2017). We argue this issue is aggravated when adapting pre-trained VLMs to robotics, as the conventional fine-tuning process for predicting dense, continuously parameterized action trajectories (or "action chunks") creates a substantial "task conflict." Specifically, the low-level, high-frequency control signals of dense trajectories represent a significant departure from the VLM's original pre-training objectives (e.g., text generation, image captioning). This misalignment can cause the model to overwrite previously consolidated competencies during fine-tuning on a new task, thereby degrading prior-task performance.

The challenges associated with this fragility are rooted in the prevailing training paradigm. Fine-tuning on voluminous, dense trajectories is not only computationally expensive but also underutilizes the high-level semantic abstraction and reasoning capabilities latent in Vision-Language Models (VLMs) (Alayrac et al., 2022; Reed et al., 2022; Hu et al., 2022). To address these issues, we introduce the Narrowing of Trajectory VLA (NoTVLA) framework (Figure 1), which rethinks the granularity of action representation. The central premise is that by narrowing supervision to sparse, semantically aligned keyframes, we can reframe the action prediction task to be more congruent

with the VLM's pre-training. This approach mitigates catastrophic forgetting by reducing the "task conflict," allowing the model to leverage its existing reasoning abilities rather than overwriting them, all while lowering the adaptation cost.

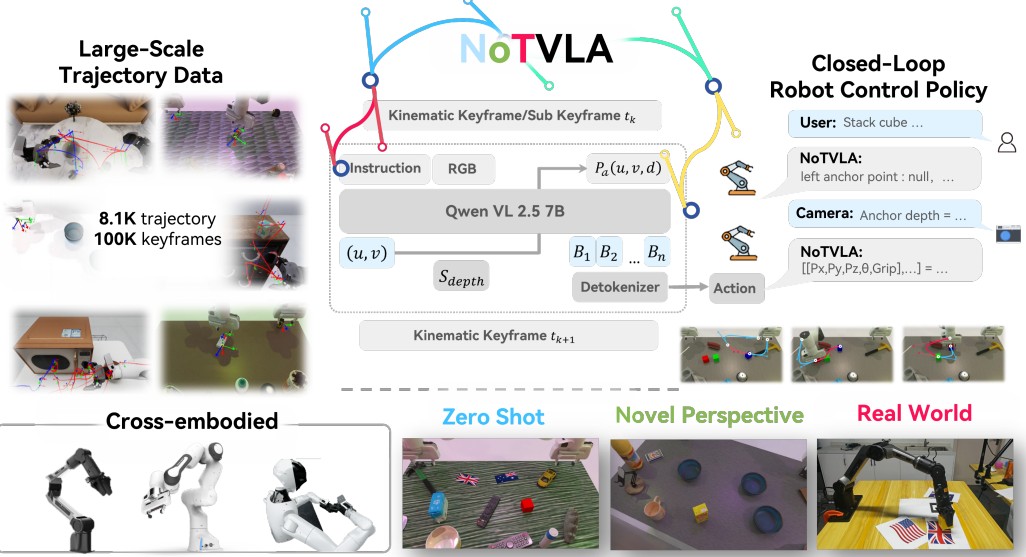

Figure 1: Overview of the NoTVLA framework. To mitigate catastrophic forgetting, NoTVLA reframes action generation from predicting dense trajectories to predicting sparse, semantically aligned keyframes, reducing task conflict with VLM pre-training. Given an instruction l and an RGB image I, the model predicts a 2D anchor point $(u_a, v_a)$ and queries its depth $d_a$ from a sensor or a depth prediction model $S_{depth}$. This depth-augmented anchor $a = (u_a, v_a, d_a)$ conditions the generation of discrete action tokens $B_i = (\mathcal{D}_i, \mathcal{U}_i, \mathcal{G}_i, \mathcal{R}_i)$, which represent depth, image coordinates, gripper state, and pose. A spline-based detokenizer then converts these tokens into a smooth, high-frequency trajectory for closed-loop robot control. The framework generalizes across robot embodiments and tasks, supporting zero-shot execution and real-world deployment.

A key innovation of the proposed framework is a trajectory planning and abstraction strategy that shifts focus from dense object-centric rollouts to a temporally compressed, spatially pruned representation of the robot end-effector path. Through temporal compression and spatial reasoning-based pruning, the method yields leaner yet decision-sufficient action sequences, enabling more sample- and compute-efficient refinement (Li et al., 2023; Liu et al., 2023b). Training on these sparse trajectories—rather than on full-resolution dense streams—improves retention and cross-task transfer. Empirically, the approach attains superior multi-task generalization relative to contemporary VLA baselines (e.g., $\pi_0$) while using over an order of magnitude less compute and no wrist-mounted camera, yet approaching the operational accuracy of specialized single-task experts.

Crucially, the framework preserves intrinsic language and reasoning capabilities of the underlying VLM, supporting zero-shot adaptation in selected novel settings (Brohan et al., 2023; Driess et al., 2023). It further enables unified deployment across heterogeneous robot embodiments and sustains competence under viewpoint shifts, indicating robustness to perceptual distributional variation.

In summary, our contributions are as follows:

- We decouple the high-level VLM from low-level action experts, achieving higher embodied-task success while substantially reducing finetuning compute.
- We show that sparse, semantically pruned trajectory supervision enhances both cross-embodiment and cross-task generalization, alleviating catastrophic forgetting.
- Unlike conventional tightly coupled designs, our framework preserves inherent vision-language reasoning, enabling complex instruction following and multi-turn interaction without degradation.

## 2 RELATED WORKS

### 2.1 SPATIAL REASONING FOR GENERALIZABLE MANIPULATION

Effective manipulation requires robust 3D spatial reasoning. One line of research embeds this capability by fine-tuning models on 3D-centric datasets (Wu et al., 2025a; Cai et al., 2024; Liu et al., 2025) or by directly processing dense 3D inputs like point clouds (Zhen et al., 2024; Yang et al., 2025b; Song et al., 2025c), which can be computationally intensive and limit modularity. Another direction provides controllers with explicit geometric guidance like keypoints (Manuelli et al., 2019; Yuan et al., 2024) or affordances (Wu et al., 2025b). Our work advances this latter principle through a decoupled **anchor-based depth inference** mechanism. Instead of requiring the VLA to process full 3D scenes, we use a lightweight head to predict a 2D anchor, inject depth from an external source, and then generate actions. This factorized design significantly reduces the model's perceptual burden and enhances modularity, allowing for flexible integration across different robot platforms and sensor configurations.

### 2.2 HIERARCHICAL ARCHITECTURES AND INTERMEDIATE REPRESENTATIONS

To manage the complexity of end-to-end models, hierarchical frameworks that decouple a high-level planner from a low-level action expert have become prominent (Black et al., 2410; Intelligence, 2025; Bjorck et al., 2025). The efficacy of this architecture hinges on the intermediate representation connecting the two modules. While prior works have used abstract sub-goals (Li et al., 2024a; de Bakker et al., 2025) or latent features (Li et al., 2025) or explicit trajectory (Yang et al., 2025a), their semantic ambiguity can burden the action expert. Our framework uses explicit 3D waypoints as a clear interface, but more importantly, we introduce a novel **spline-based action detokenizer** to bridge the frequency gap. This module translates low-frequency waypoints from the VLA into a smooth, high-rate, and jitter-free robot trajectory online, directly addressing a core challenge in deploying hierarchical systems.

### 2.3 FROM DENSE TO SPARSE ACTION REPRESENTATIONS IN VLA MODELS

Recent Vision-Language-Action (VLA) models have made significant strides by mapping multimodal prompts to dense, high-frequency action sequences (Brohan et al., 2022; 2023; Kim et al., 2024; Team et al., 2024; Song et al., 2025c; Li et al., 2025; Wen et al., 2025; Zhao et al., 2025; Song et al., 2025b;a; Fan et al., 2025; Wang et al., 2025). A lot of datasets (Khazatsky et al., 2025; Mu et al., 2021; Walke et al., 2023; Liu et al., 2023a; James et al., 2019; AgiBot-World-Contributors et al., 2025; Mu et al., 2025; Chen et al., 2025b) are built under this VLA action pattern. However, this end-to-end paradigm's reliance on dense supervision imposes immense data requirements and can lead to challenges like catastrophic forgetting (French, 1999). This has spurred a shift towards sparse action representations, which offer a more tractable interface between planning and control. While methods like uniform downsampling risk losing critical motion details, and learned segmentation models add complexity, our approach introduces a **kinematics-based keyframe selection** protocol. By identifying keyframes based on physical properties like acceleration and gripper state changes, we achieve a meaningful and efficient trajectory sparsification that preserves task-critical information without requiring additional learned components.

## 3 METHOD

As shown in Figure 2 we present a VLA framework that combines the hardware operability of classical controllers with the generalization ability of VLMs under a unified, cross-robot trajectory annotation. The system consists of: (1) anchor-based depth inference that equips VLMs with efficient and scalable depth reasoning, (2) a spline-based action detokenizer that turns discrete tokens into smooth, high-rate trajectories, and (3) kinematics-based keyframe selection to preserve task-critical motion. During training, we employ a kinematics-based keyframe selection method (Section 3.2) to provide sparse supervision for our VLA model, using only the most critical end-effector poses. At inference time, the procedure unfolds in two stages. First, the model generates a planned trajectory via anchor-based depth inference—a two-step, text-guided prediction detailed in Section 3.1. Sub-

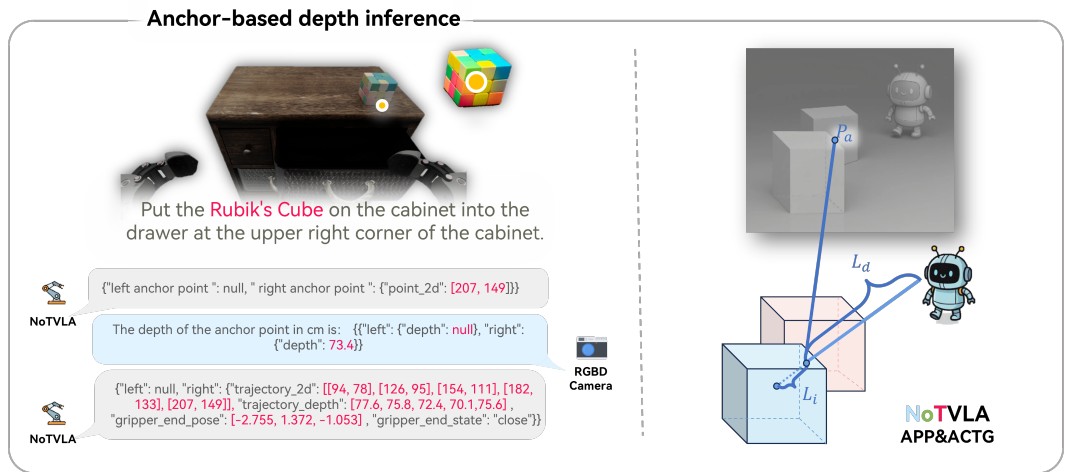

Figure 2: "Anchor Point Prediction (APP) and Anchor-Conditioned Token Generation (ACTG). Duringinference, APP predicts a 2D anchor from the image and language instruction. The anchor depth is obtained from an external sensor, and ACTG generates action tokens conditioned on the depth-augmented anchor. These tokens are converted into smooth trajectories via the detokenizer."

sequently, this trajectory is processed by a spline-based action detokenizer to produce a sequence of smooth, executable actions.

### 3.1 ANCHOR-BASED DEPTH INFERENCE

We turn absolute depth prediction into anchor-conditioned reasoning: a 2D anchor localizes the task-relevant region; an external depth source supplies its depth; the VLA then generates a full multi-modal action token stream conditioned on this tri-modal cue (image, language, anchor-with-depth). This decoupling lowers vision model burden and improves cross-view robustness.

**Anchor Point Prediction (APP).** At *inference time* APP receives only the RGB image and language instruction and outputs a *single* 2D anchor (being the projection of the 3D pose of the gripper end effector in the key frame onto the image plane):

$$(u_a, v_a) = \mathcal{M}(\mathbf{I}, \mathbf{l}), \qquad (u_a, v_a) \in [0, W] \times [0, H]. \tag{1}$$

Training can supervise this point as (i) the object center, (ii) the first contact pixel, or (iii) a task-specific salient location. No depth is produced inside APP. The anchor depth is then obtained from an *external* source:

$$d_a = \mathcal{S}_{\text{depth}}(\mathbf{I}, u_a, v_a), \tag{2}$$

where $\mathcal{S}_{\text{depth}}$ can be: (a) an RGB-D or LiDAR sensor ($d_a = D(u_a, v_a)$) or (b) a monocular estimator. The final depth-augmented anchor is $a = (u_a, v_a, d_a)$.

**Anchor-Conditioned Token Generation (ACTG).** ACTG is a training method that inputs the depth-augmented anchor $\text{ANCHOR}(u_a, v_a, d_a)$ to enhance the ability of 3D inference of VLM. At inference it consumes $(\mathbf{I}, \mathbf{l}, a)$ and autoregressively outputs a trajectory token sequence containing depth, image-plane coordinates, gripper, and pose modalities: depth $\mathcal{D} = \{d_i\}$; image UV $\mathcal{U} = \{(u_i, v_i)\}$; gripper state $\mathcal{G} = \{g_i\}$; gripper pose $\mathcal{R} = \{r_i\}$ with Euler angles $r_i = [\theta_{xi}, \theta_{yi}, \theta_{zi}]^\top$, which is serialized as

$$\mathbf{S} = [\text{CLS}, \text{IMG}, \text{TXT}, \text{ANCHOR}(u_a, v_a, d_a), B_1, \ldots, B_N, \text{EOS}]. \tag{3}$$

$$B_i = (\mathcal{D}_i, \mathcal{U}_i, \mathcal{G}_i, \mathcal{R}_i). \tag{4}$$

Reconstructed 3D waypoints use camera intrinsics:

$$\mathbf{p}_i = \tilde{d}_i \, \mathbf{K}^{-1} [\tilde{u}_i, \ \tilde{v}_i, \ 1]^\top. \tag{5}$$

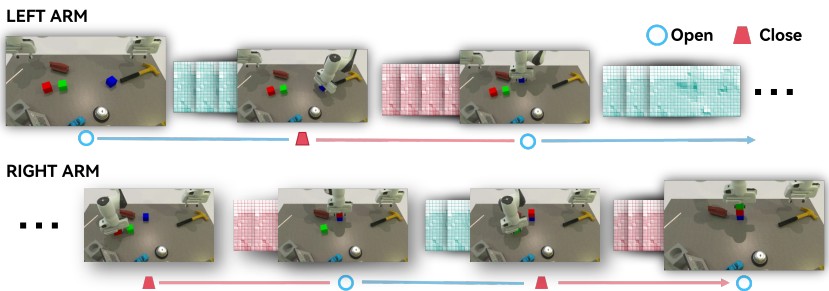

Figure 3: Keyframe selection based on gripper posture. The images show the gripper's movements over time, with keyframes selected based on the open/close states of the left and right arms. The blue and red lines represent the gripper's trajectory, with the corresponding open and close states annotated, highlighting the transitions between key poses during object manipulation.

The sequence $\mathbf{S}$ starts at the current frame and ends at another keyframe (Sec. 3.2). Optionally, depths (and/or UV) can be modeled relative to the anchor: $d_i = d_a + \Delta d_i$ with the same quantization applied to $\Delta d_i$ over a symmetric range; we keep absolute depth for simplicity.

### 3.2 KINEMATICS-BASED KEYFRAME SELECTION

We segment raw demonstrations into logical phases (Chen et al., 2025a), as shown in Figure 3. Let the end-effector pose be $\boldsymbol{T}(t) = [x(t), y(t), z(t), \theta_x(t), \theta_y(t), \theta_z(t)]^\top$ and the gripper state $G(t) \in \{0, 1\}$. A keyframe at time $t_k$ satisfies

$$\left\|\ddot{\boldsymbol{T}}(t_k)\right\|_2 > \alpha \ \lor \ \lim_{\epsilon \to 0^+} G(t_k - \epsilon) \neq \lim_{\epsilon \to 0^+} G(t_k + \epsilon), \tag{6}$$

where $\alpha$ is an threshold. In practice, accelerations are computed via simple finite differences, and the threshold (and any component weights) are hand-designed per task; no additional filtering or post-processing is applied.

We discretize the motion between consecutive keyframes and express all poses in the camera frame following the RoboTwin convention (Mu et al., 2024). To reduce information loss, we insert sub-keyframes (Pertsch et al., 2020) and uniformly downsample each segment with constant $N$ and equal temporal spacing:

$$\boldsymbol{T}_{\text{down}} = \{\boldsymbol{T}_i\}_{i=0}^{N-1} = \text{UniformSample}(\boldsymbol{T}_k, \boldsymbol{T}_{k+1}, N). \tag{7}$$

This preserves temporal coherence while lowering compute during training.

### 3.3 SPLINE-BASED ACTION DETOKENIZER

Robots require smooth, millisecond-level control, whereas VLMs emit tokens at low rates ($\sim$1–2 Hz). We bridge this gap with interpolation and closed-loop updates (Miura et al., 2024; Urain et al., 2022). The token sequence $\{B_i\}$ produced by ACTG, consisting of waypoints in the camera frame, is first transformed into the robot's base (world) frame. Let $\boldsymbol{T}_c^w$ be the transformation from camera to world coordinates. Each waypoint $\boldsymbol{p}_{c,i}$ is converted to the world frame:

$$\boldsymbol{p}_{w,i} = \boldsymbol{T}_c^w \boldsymbol{p}_{c,i}. \tag{8}$$

The resulting 3D waypoints $\{\boldsymbol{p}_{w,i}\}$ are then used for trajectory generation. We employ cubic spline (Farin, 2002) interpolation for the XYZ position to ensure a smooth path:

$$\boldsymbol{P}(t) = \text{Spline}(\{\boldsymbol{p}_{w,i}\}, t). \tag{9}$$

For orientation, the Euler angles $\{\boldsymbol{r}_i\}$ are first converted to quaternions $\{\boldsymbol{q}_i\}$. We then use Spherical Linear Interpolation (SLERP (Shoemake, 1985)) between consecutive quaternions to generate a smooth and unambiguous rotation trajectory:

$$\boldsymbol{q}(t) = \text{SLERP}(\boldsymbol{q}_i, \boldsymbol{q}_{i+1}, \frac{t - t_i}{t_{i+1} - t_i}), \quad t \in [t_i, t_{i+1}]. \tag{10}$$

This combined approach ensures a smooth high-frequency trajectory in both position and orientation, suitable for robot execution.

However, this procedure will cause an error on the trajectory. A cubic B-spline reconstruction (Farin, 2002; Piegl & Tiller, 1997; de Boor, 2001)

$$\hat{\boldsymbol{T}}(t) = \sum_{j=0}^{N} B_{j,3}(s(t))\, \boldsymbol{T}_j \qquad (11)$$

exhibits error bounded by

$$\Delta \boldsymbol{T}_{\text{sub}}(t) \le K \, \max \|\boldsymbol{T}^{(4)}(t)\|_2 \left(\tfrac{\Delta t}{M+1}\right)^4, \qquad (12)$$

indicating sub-keyframes near keyframes sharply reduces error. In practice, we sample at least 10 sub-keyframes between keyframes, which makes the inference close to keyframe very close to gt. The design deploys on Franka Reach, Aloha–AgileX, PiPER, and AGIBOT G1 with closed-loop stability. The details are in the Appendix.

# 4 EXPERIMENTS

## 4.1 COMPARISON WITH EXISTING METHODS

### 4.1.1 COMPARISON OF SUCCESS RATES

In a constrained sensing setup using only head-mounted monocular RGBD and trained across multiple tasks, our model attains accuracy close to single-task experts (e.g., ACT (Zhao et al., 2022), DP (Chi et al., 2023), DP3 (Li et al., 2024b)) and classical VLA (e.g., $\pi_0$) as shown in Table 1, shows clear advantages on longer-horizon tasks due to preserved VLM reasoning, but this strength is underrepresented by short-horizon benchmarks and limited when relevant objects fall outside the image.

Table 1: Performance comparison between expert models and generalist models on various robotic tasks in RoboTwin 2.0 (Mu et al., 2025). Full table is in the Appendix E.1

| Task Name | Expert Models | | | Generalist Models | | |
|---|---|---|---|---|---|---|
| | ACT | DP | DP3 | $\pi_0$ | RDT | Ours |
| click alarmclock | 0.32 | 0.61 | 0.77 | 0.63 | 0.58 | **0.78** |
| click bell | 0.58 | 0.54 | 0.90 | 0.44 | 0.18 | **0.94** |
| grab roller | 0.94 | **0.98** | **0.98** | 0.96 | 0.50 | 0.97 |
| handover mic | 0.85 | 0.53 | **1.00** | 0.98 | 0.13 | **0.99** |
| move playingcard away | 0.36 | 0.47 | 0.68 | 0.53 | 0.03 | **0.77** |
| open laptop | 0.56 | 0.49 | 0.82 | **0.85** | 0.15 | 0.57 |
| press stapler | 0.31 | 0.06 | 0.69 | 0.62 | 0.34 | **0.94** |
| Shake bottle | 0.74 | 0.65 | **0.98** | 0.97 | 0.31 | **0.94** |

In the *AGIBOT Challenge*, our code was evaluated by the official server and demonstrated clear improvements over the UniVLA baseline. Overall, our approach outperformed the baseline across multiple tasks, with particularly pronounced gains on *Open drawer and store items* and *Pickup items from the freezer*, and these advantages were confirmed in the official evaluation as shown in Table 2.

Table 2: AGIBOT Challenge Official Test Results. Columns 1-10 correspond to the 10 official tasks in the challenge. Details in the Appendix A.5.1.

| Method | Total | 1 | 2 | 3 | 4 | 5 | 6 | 7 | 8 | 9 | 10 |
|---|---|---|---|---|---|---|---|---|---|---|---|
| UniVLA | 2.795 | 0.097 | 0.020 | 0.033 | 0.350 | 0.260 | 0.400 | 1.000 | 0.080 | 0.375 | 0.180 |
| Ours | 3.697 | 0.161 | 0.304 | 0.350 | 0.320 | 0.448 | 0.510 | 0.600 | 0.262 | 0.350 | 0.392 |

On the other hand, as we can see in Figure 4, we use far fewer training steps than many alternatives and avoid additional single-task fine-tuning, which demonstrates reliable multitask capability. This efficiency also leaves headroom for scaling while maintaining strong language competency. Compared with other VLA models such as $\pi_0$, our approach thus offers a more scalable solution with efficient language processing. Although our setting does not require single-task fine-tuning, it remains a technique from which prior VLA paradigms can still benefit. Together, these design choices deliver higher language-processing efficiency and a cleaner scaling path while maintaining strong language competency.

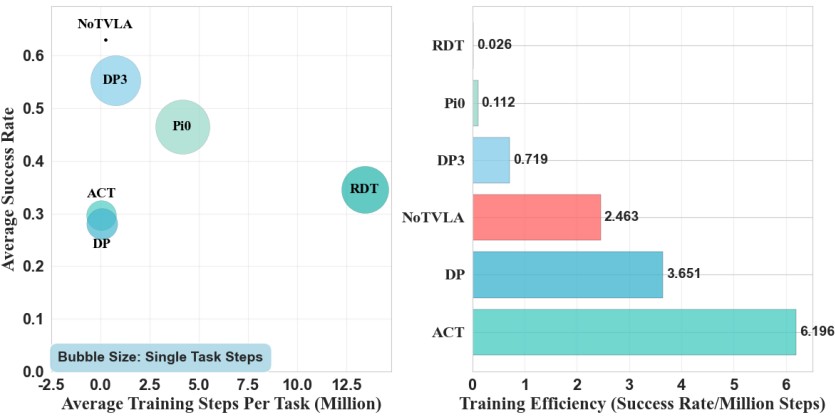

Figure 4: Training steps and average success rate of different works. NoTVLA uses 7B weight, close to RDT and $\pi_0$. The bubble size refer to the single task training step of different works. NoTVLA without single task training still performs better than other models.

Building on this efficiency perspective, we further examine the training cost and fine-tuning requirements. More details of different steps are in the Appendix C.1 No additional single-task fine-tuning was required, as the marginal gains proved limited. This design also brought a substantial reduction in training cost: for instance, training on Robotwin required only 32 GPU hours, and even with the compute budget reduced to roughly 8 GPUs, the model already demonstrated execution capability on most tasks (see ablation study for details). To contextualize efficiency, the left panel of the bubble chart summarizes total training steps, single-task fine-tuning steps, and efficiency by steps: the x-axis denotes overall training steps, bubble size represents single-task fine-tuning steps (zero for NoTVLA), and the y-axis shows success rate under the given budgets. For comparison, ACT, DP, and DP3 are single-task expert models, whose total steps scale linearly with the number of tasks, while $\pi_0$ and RDT are general-task models that combine full-task fine-tuning with per-task fine-tuning proportional to the task count.

### 4.1.2 ZERO-SHOT COMPARISON

Our method demonstrates strong generalization, performing well in Robotwin's hard scenarios, including unseen instructions and backgrounds. It responds to color cues absent from training and handles task descriptions that are the inverse of those seen during training. Additionally, it exhibits zero-shot transfer to abstract concepts (e.g., flags) not included in the training set. The results are in Table 3.

For example, all training instances of *stack block two* use a green-over-red ordering. At test time, we invert the instruction (stack red on green) and introduce new colors and descriptions. Our model maintains strong performance under these changes, while other models fail to show the same zero-shot robustness.

To fairly compare zero-shot generalization, we collect pre-training and fine-tuning data from Robotwin and train both the open-source $\pi_0$ and our model. To avoid overtraining as a cause of weak generalization, we evaluate two $\pi_0$ checkpoints with different training budgets, both of which perform well on *move pillbottle pad* and *stack blocks two*. We then test these $\pi_0$ checkpoints and our model on zero-shot tasks, inverting instruction semantics to probe adversarial language and

replacing block colors to test understanding of more complex descriptions. Compared to $\pi_0$, our model remains responsive to adversarial instructions and complex language. $\pi_0$, while strong on single tasks, shows limited interaction flexibility and suffers from catastrophic forgetting when scaling to multiple tasks. $\pi_0$'s subtask decomposition also assumes strictly defined subtasks, risking the loss of crucial linguistic details (e.g., color), which constrains compositionality. For low-level skills, complex instruction combinations are often necessary, and atomic subtasks reduce coverage of diverse scenarios. As for different models, they also have different performances. We have tried Qwen2.5-VL 7B, 3B and InterVL 2.5 1B. Details are shown in the Appendix A.3.

Table 3: Performance comparison of tasks in the training set and zeroshot tasks. Details of trained tasks are in the Appendix. Details of zero-shot tasks are in the Appendix A.5.3

| Task Name | $\pi_0$-30000 | $\pi_0$-5000 | NoTVLA 7B |
|---|---|---|---|
| **Trained Tasks** | | | |
| place mouse pad | 23% | 22% | 23% |
| stack blocks two | 90% | 65% | 87% |
| **Zero-shot Tasks** | | | |
| stack blocks two (inverse) | 0.0% | 0.0% | 70% |
| stack random color blocks | 30% | 18% | 69% |
| move block random color pad | 10% | 9% | 18% |
| stack numberblocks two | 0.0% | 0.0% | 58% |
| stack random blocks two from three | 1.0% | 0.0% | 57% |
| place block flagpad | 32% | 55% | 78% |
| place block flagpad (choice) | 25% | 56% | 62% |

### 4.1.3 Trajectory Planning Comparison

We validate our framework by comparing its trajectory generation to the planning-based Magma planning trajectory shape across geometric and temporal metrics in the OOD field trajectory in RoboTwin, and it outperforms on all: higher coverage (F1 0.955 vs 0.928), markedly lower DTW (0.456 vs 0.835) and Fréchet distance (0.104 vs 0.230), reduced start/end errors by 46%/59%, consistently smaller orthogonal distances, and favorable Hausdorff results. These gains show our spline-based detokenizer produces more accurate, consistent trajectories, effectively bridging low-frequency model outputs to high-frequency robot control. Full comparison is in the Appendix C.2.

## 4.2 Ablations and Analyses

### 4.2.1 Depth Reasoning Ablation

In NoTVLA, **Anchor Point Prediction** (APP) and **Anchor-Conditioned Token Generation** (ACTG) are introduced to enhance spatial reasoning. APP instructs the model to select a perceptual anchor within the image and query the depth at that specific point, while ACTG provides a single-point depth token to serve as the foundation for subsequent reasoning. A central question arises: do explicit depth inputs, as opposed to monocular depth inference performed by the Vision-Language Model (VLM), improve performance in manipulation tasks?

To address this, we evaluate three distinct variants across a range of representative tasks, including spatiotemporal semantics, object recognition, and long-horizon sequences:
(1) **w.APP w.ACTG**, which first outputs an anchor point and acquires the associated depth.
(2) **w.APP w/o.ACTG**, which outputs a anchor point but lacks depth from the sensor.
(3) **w/o.APP w/o.ACTG**, which performs direct reasoning without explicit anchor point and depth information.

Our results that are shown in Table 4 demonstrate that the inclusion of explicit depth inputs significantly improves success rates, particularly for **long-horizon tasks**. These tasks involve broader depth ranges and intraclass scale variation, both of which induce perspective distortion that complicates monocular depth estimation. In addition, such tasks often require multiple rounds of spatial reasoning, where reference points play a critical role in stabilizing repeated inferences. In particu-

Table 4: Performance comparison of tasks across different methods.

| Task Name | NoTVLA w. APP w. ACTG | NoTVLA w/o APP w/o ACTG | NoTVLA w. APP w/o ACTG |
|---|---|---|---|
| **Object Recognition** | | | |
| place a2b left | 0.38 | 0.28 | 0.30 |
| place a2b right | 0.34 | 0.24 | 0.32 |
| place object scale | 0.43 | 0.39 | 0.36 |
| **Long-term Tasks** | | | |
| blocks ranking size | 0.53 | 0.33 | 0.28 |
| blocks ranking rgb | 0.76 | 0.66 | 0.59 |
| Stack block three | 0.49 | 0.32 | 0.30 |
| Stack bowl three | 0.83 | 0.78 | 0.68 |

Table 5: Performance on multi-perspectives on different tasks.

| Task Name | In Domain | Out of Domain |
|---|---|---|
| **Object Recognition** | | |
| place a2b left | 0.27 | 0.28 |
| place a2b right | 0.27 | 0.30 |
| place object scale | 0.49 | 0.45 |
| **Long-term Tasks** | | |
| Stack block three | 0.35 | 0.31 |
| Stack bowl three | 0.52 | 0.55 |
| blocks ranking size | 0.19 | 0.12 |
| blocks ranking rgb | 0.50 | 0.45 |

lar, APP alone further enhances performance in cespatio-temporalemporal tasks, likely due to the proximity of the predicted anchor to the manipulation locus, providing a consistent reference for subsequent reasoning.

### 4.2.2 GENERALIZATION OF NOVEL PERSPECTIVES

Table 5 compares task performance between *In Domain* (ID) camera views seen during training and *Out of Domain* (OOD) novel views. The metric is the success rate for each task under these two conditions. The results demonstrate that our model maintains strong performance even when faced with novel perspectives, with only a minor drop in success rate for most tasks (e.g., *Stack block three* drops from 0.35 to 0.31; *blocks ranking rgb* from 0.50 to 0.45). As described in the method section, we decouple 3D reasoning from direct depth map prediction and all the VLM inference are in the camera perspective. This approach avoids forcing the VLM to learn a dense, view-dependent action representation. Instead, it learns to reason about task-relevant 3D geometry relative to a single, robustly localized anchor. This decoupling significantly enhances the model's ability to generalize across different camera viewpoints, as the core reasoning process is less sensitive to changes in perspective.

### 4.2.3 PERFORMANCE OF DIFFERENT TRAINING STEPS

The task success rate, evaluated at different training steps (from 500 to 4000), was on mutlitask in the Appendix C.1. The tasks are categorized into short, medium, and long horizons, representing increasing levels of complexity and action sequence length. Our experiments shows that performance does not uniformly increase with more training steps; instead, it often fluctuates. For instance, the *grab roller* task achieves a perfect success rate at only 500 steps but declines thereafter, suggesting potential overfitting. Similarly, other tasks like *place empty cup* and *place bread basket* exhibit non-monotonic performance curves. On the other hands, task horizon significantly impacts performance. Short-horizon tasks like *grab roller* and *place empty cup* achieve high success rates early in training, as they involve simpler motion sequences. In contrast, long-horizon tasks such as *stack blocks three* start with low success rates and require more training to show improvement. This is because our method relies on **Kinematics-Based Keyframe Selection** to segment demonstrations. Long-horizon tasks has more keyframes, making the sequence modeling problem for the VLM more challenging.

## 5 CONCLUSION

We present a VLA framework that bridges VLM high-level reasoning with precise low-level control via three components—anchor-based depth inference for simplified 3D perception, kinematics-driven keyframe selection, and a spline-based detokenizer for smooth, high-frequency trajectories. Across diverse manipulation tasks, it matches or surpasses expert and generalist baselines, trains efficiently with fewer steps and no task-specific finetuning, and generalizes zero-shot to novel instructions, objects, and scenarios. Limitations include sensitivity to objects outside the camera's field of view and benchmarks that underrepresent long-horizon reasoning; future work will enhance sensing and evaluate on more complex, long-horizon tasks toward scalable, efficient generalist robots.

## ETHICS STATEMENT

This work adheres to the ICLR Code of Ethics. All experiments involving our physical robot were conducted in a controlled laboratory with strict safety protocols, including emergency stops and prior validation in simulation. Our study did not involve human participants or sensitive data. We acknowledge the dual-use nature of robotics research and affirm that our work is intended solely for beneficial applications, such as industrial and assistive automation.

## REPRODUCIBILITY STATEMENT

We have taken substantial steps to ensure our research is reproducible. The core methodology of our proposed approach is described in detail in Section 3. To facilitate the verification of our results, comprehensive details regarding the datasets, including preprocessing steps, are provided in Appendix A.1. Furthermore, the complete experimental setup is outlined in Appendix A.2, while the specific training procedures and hyperparameters for each task are detailed in Appendix C.1. We are committed to releasing our full source code and pre-trained models in a public repository upon acceptance of this paper.

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

USE OF LLMS

We used large language models (LLMs) only for minor assistance in polishing the language and adjusting the presentation of tables. No LLMs were involved in designing the methodology, conducting experiments, or analyzing results.

# A  DATA AND IMPLEMENTATION DETAILS

## A.1  DATA SOURCES AND COVERAGE

We integrate three data sources—ManiSkill, RoboTwin 2.0, and AgiBot World—within a unified framework to create a cross-ontology, cross-view, and cross-scenario manipulation corpus.

**ManiSkill**: Provides high-quality, large-scale single-task trajectory data, including stack cube, push cube, and pull cube tool tasks, with a total of 3000 trajectories.

**RoboTwin 2.0**: Focuses on dual-arm manipulation and strong domain randomization (texture/lighting/table height/clutter/linguistic perturbation), including multiple robots such as Aloha-AgileX, ARX-X5, Franka, Piper, UR5, and others. We use a high-quality subset of 40 tasks, totaling around 2000 trajectories, divided according to the official evaluation. Extensive data from various perspectives were collected, with precise camera intrinsic and extrinsic parameters. We also created 30 custom tasks based on the official scenarios, collecting a total of 200 trajectories for 10 tasks.

**AgiBot World**: Contains multi-modal sequences from near-realistic collection processes (AgiBot G1), including RGB-D, fisheye, multi-view calibration, and step-level language. To ensure stable simulation-to-real transitions, we use sub-scenes resembling home/retail/office environments, totaling 10 tasks and 500 trajectories. The evaluation uses official training data from the AgiBot challenge, and the evaluation is performed on their official servers, with code reviewed by the AGIBOT team.

**Private Robot Dataset and Open Source Robot Dataset**: We collected 1k+ trajectories on Franka and PiPER robot in custom robot setting and followed task configuration of RoboTwin dataset. AGIBOT G1 dataset was from AGIBOT official.

## A.2  EXPERIMENTAL SETUP

**Usage.** We train **NoTVLA** on a unified simulation mix: 2000 trajectories from RoboTwin (40 tasks), 3000 from ManiSkill (3 tasks), and 1000 trajectories from AGIBOT (10 tasks). Each trajectory is normalized to the camera frame and discretized at gripper state changes; at each keyframe we store image-plane UVs, depth tokens, gripper pose, and a binary gripper flag. Logs are serialized as JSON+NPZ with timestamps and camera intrinsics/extrinsics. The control target is the end effector across embodiments, aligning gripper coordinates to the fingertip midpoint; joint angles are retained for diagnostics. Embodiments include Aloha-AgileX, ARX-X5, Franka Panda, UR5, Piper, and AgiBot-G1; inference uses head cameras, with wrist views used only for augmentation. We sample 10 frames per segment (upsampling to 10 via splines) and allocate 50/0/100 for train/val/test with instruction de-duplication. Language consists of 100 native prompts plus 20 crowd-sourced rewrites per task. All data are versioned with checksums and visually inspected to filter low-quality samples.

**Datasets.** We use three complementary sources: **RoboTwin 2.0**, **AgiBot World** and self collect dataset. RoboTwin2.0 contributes large-scale bimanual demonstrations with domain randomization over textures, lighting, table height, clutter, and language prompts; we adopt its evaluation split and dual-arm embodiments (Aloha-AgileX, ARX-X5, Franka, Piper, UR5). AgiBot World offers real-to-sim–friendly sequences from dual-arm humanoids (AgiBot G1), with RGB-D and fisheye streams, calibration, and step-level language. Across sources, trajectories bundle synchronized visual frames, depth, proprioception, end-effector pose, and gripper state, together with the instruction text. This mixture covers short- and long-horizon skills (e.g., opening/closing, stacking, placement, tool use) in household, retail, and office scenes. In combination, the corpora provide complementary coverage: ManiSkill contributes controllable 3D variety, RoboTwin supplies structured bimanual tasks with broad randomization, and AgiBot contributes high-fidelity, real-world diversity.-

### A.3 TRAINING WEIGHT OF TOKENS

During inference, token types contribute differently to model accuracy, which is shown in Table 6. One important consideration is whether increasing the loss weight of trajectory tokens improves execution. Under NoTVLA training, scaling the trajectory-token weight by an order of magnitude does *not* improve manipulation. In fact, it substantially degrades language proficiency and reasoning. After reweighting, we observe declines on standard language benchmarks and lower task accuracy in simulation. A likely explanation is overfitting: the model over-emphasizes numeric trajectory reasoning at the expense of logical interpretation and analysis of instructions.

### A.4 TRAINING DETAILS

**RDT.** RDT in RoboTwin 2.0 evaluation was pretrained for 100,000 steps with a batch size of 16 per GPU on 8 GPUs, and all single-task fine-tuning was conducted for 10,000 steps with a batch size of 16 per GPU on 4 GPUs.

**Pi0.** Pi0 in RoboTwin 2.0 evaluation was pretrained for 100,000 steps with a batch size of 32, and all fine-tuning was performed for 30,000 steps using the same batch size.

**ACT.** ACT in RoboTwin 2.0 evaluation was trained under a unified setup with a chunk size of 50, batch size of 8, and single-GPU training for 6,000 epochs.

**DP.** DP in RoboTwin 2.0 evaluation was trained for 600 epochs with a batch size of 128 and a planning horizon of 8.

**DP3.** DP3 in RoboTwin 2.0 evaluation was trained for 3,000 epochs with a batch size of 256, using a planning horizon of 8 and a point cloud resolution of 1,024, with precise segmentation of the background and tabletop.

**NoTVLA.** NoTVLA in RoboTwin 2.0 evaluation was trained for 2,000 epochs with a batch size of 128.

| Model | STEM | Social | Hum. | Other | Avg. |
|---|---|---|---|---|---|
| **MMLU** | | | | | |
| Qwen7B | 65.64 | 79.79 | 63.27 | 74.89 | 70.08 |
| Ours w/ $10\times$ | 36.48 | 51.48 | 37.79 | 43.34 | 41.79 |
| Ours | 55.30 | 74.52 | 53.75 | 67.00 | 61.69 |
| **CMMLU** | | | | | |
| Qwen7B | 71.36 | 75.63 | 77.30 | 75.84 | 75.11 |
| Ours w/ $10\times$ | 35.12 | 44.33 | 43.11 | 42.03 | 41.48 |
| Ours | 57.17 | 66.70 | 67.10 | 63.44 | 63.88 |
| **CEVAL** | | | | | |
| Qwen7B | 73.02 | 84.00 | 76.26 | 73.96 | 76.15 |
| Ours w/ $10\times$ | 38.14 | 47.27 | 48.64 | 44.01 | 43.68 |
| Ours | 58.14 | 75.27 | 64.59 | 60.16 | 63.45 |

Table 6: Performance on MMLU, CMMLU, and CEVAL (STEM, Social Sciences, Humanities, Other, Avg.).

### A.5 TASKS DETAILS DESCRIPTION

#### A.5.1 AGIBOT CHALLENGE

**Clear the countertop waste - task 1** Remove loose trash and food scraps from the countertop, place them into the appropriate waste or recycling container, and wipe the surface clean.

**Open drawer and store items - task 2** Open the specified drawer, place the target items inside following organizational guidelines, arrange them neatly, and close the drawer securely.

**Heat the food in the microwave - task 3** Place the food in a microwave-safe container, set the appropriate power and time, start the microwave, and carefully remove and check the food temperature when finished.

**Pack moving objects from conveyor - task 4** Identify and pick moving objects from the conveyor belt, place them into designated packing boxes or trays while maintaining speed and stability.

**Pickup items from the freezer - task 5** Open the freezer, locate and retrieve specified frozen items using proper protection (e.g., gloves), move them to the target location, and close the freezer.

**Restock supermarket items - task 6** Retrieve products from stock, refill shelves according to merchandising rules (facing front, orderly arrangement), replace expired or mislabeled items as needed.

**Pack in the supermarket - task 7** Pack purchased goods for a customer by grouping items sensibly, protecting fragile goods, and arranging bags or boxes for safe and easy carrying.

**Make a sandwich - task 8** Prepare bread and fillings according to the recipe, assemble ingredients in order, cut if required, and present a clean, ready-to-eat sandwich.

**Clear table in the restaurant - task 9** Remove dishes, utensils, and leftovers from the table, clear and sanitize the surface, and reset place settings if necessary for the next guest.

**Stamp the seal - task 10** Retrieve the stamp, align it with the required location on the document, press firmly to produce a clear impression, and verify alignment and clarity.

### A.5.2 ROBOTWIN OFFICIAL TASKS

**Beat Block Hammer** Use a hammer to strike the block target repeatedly until the action is completed or the block is secured.

**Blocks Ranking RGB** Rank blocks by visual RGB features and sort them according to the specified criterion.

**Blocks Ranking Size** Rank blocks by size and arrange them in the required order.

**Click Alarmclock** Locate the alarm clock and press or click the designated button to silence or activate the alarm.

**Click Bell** Press the bell actuator at the indicated position to produce a ringing signal.

**Dump Bin Bigbin** Open the large bin and empty its contents into the designated disposal area.

**Grab Roller** Reach for and grasp the roller object securely, then lift or move it to the target location.

**Handover Mic** Grasp the microphone and transfer it to another agent or specified handover position.

**Hanging Mug** Pick up a mug from a hanging position and place it on the table or target surface.

**Lift Pot** Grasp the pot handle(s) and lift it carefully to the required height or location.

**Move Can Pot** Move the can and the pot to their respective target positions as specified.

**Move Pillbottle Pad** Pick up the pill bottle and place it on the target pad or designated area.

**Move Playingcard Away** Remove the playing card from its current location and place it at the specified away location.

**Move Stapler Pad** Pick up the stapler and position it on the given pad or surface.

**Open Laptop** Open the laptop lid to a functional angle and ensure it is ready for use.

**Open Micro Wave** Open the microwave door and prepare the interior for loading or unloading items.

**Place A2B Left** Pick up object A and place it at location B on the left-side target as specified.

**Place A2B Right** Pick up object A and place it at location B on the right-side target as specified.

**Place Bread Basket** Place the bread item into the basket, arranging it neatly.

**Place Bread Skillet** Place the bread onto the skillet or pan and position it correctly for cooking or serving.

**Place Burger Fries** Pack or arrange the burger and fries together in the specified container.

**Place Can Basket** Place the can into the basket at the designated spot.

**Place Can Plasticbox** Place the can into the plastic box following placement constraints.

**Place Container Plate** Put the container onto the plate or place the plate into the container as required.

**Place Dual Shoes** Pick up the pair of shoes and place them together at the target location.

**Place Empty Cup** Place an empty cup at the specified position without spilling or tilting.

**Place Fan** Position the fan at the target location and orientation.

**Place Mouse Pad** Place the mouse pad flat at the designated workspace area.

**Place Object Basket** Place the specified object into the basket, ensuring stable placement.

**Place Object Scale** Place the object on the scale platform for weighing.

**Place Object Stand** Position the object on the stand securely and centered.

**Place Phone Stand** Place the phone onto the stand in the correct orientation.

**Place Shoe** Pick up the shoe and place it at the designated spot.

**Press Stapler** Press down the stapler mechanism to staple the target documents or materials.

**Put Object Cabinet** Place the object into the cabinet compartment and close the door if required.

**Rotate QR Code** Rotate the QR code marker to the specified orientation for scanning.

**Scan Object** Position the scanner or object and perform a scan to capture required data.

**Shake Bottle** Grasp the bottle and shake it vertically or as specified to mix contents.

**Shake Bottle Horizen** Grasp the bottle and shake it horizontally to achieve the required motion.

**Stack Block Three** Stack three blocks vertically in the specified order and ensure stability.

**Stack Block Two** Stack two blocks as required and confirm alignment.

**Stack Bowl Three** Stack three bowls together with stable nesting.

**Stack Bowl Two** Stack two bowls together with proper alignment.

**Stamp Seal** Align the seal with the document and press to produce a clear stamp impression.

**Turn Switch** Rotate or flip the switch to the target position to change device state.

### A.5.3 ROBOTWIN CUSTOM TASKS

**Stack Blocks Two Inverse** Using a previously seen instruction, stack two blocks in the inverse order. The order in instruction is inversed, also.

**Stack Random Color Blocks** Following a unseen color-order instruction, stack two random color blocks in the required order.

**Move Block Random Color Pad** Pick the colorful block and place it onto the correspondingly colored pad, matching color pairing and ensuring stable placement.

**Stack Numberblocks Two** Pick the two specified number blocks and stack them in required numerical order, bottom to top.

**stack random blocks two from three** Given three candidate blocks, randomly select any two and stack them in required order of color.

**Place Block Flagpad** Pick the target block and place it onto the flag-marked pad, centered and fully supported.

**Move Colorfulblock Colorfulpad** Pick the colorful block and place it onto the pad with the matching color pattern, aligning edges for stable contact.

**Place Block Flagpad (choice)** From multiple pads, choose a flag-labeled pad and place the block there.

**Place A2B Randomly** Pick an object from region A and place it at a random valid pose within region B, staying within bounds and avoiding obstacles.

## B    TRAJECTORY STRUCTURE AND STORAGE FORMAT

Each trajectory includes the following components:

**Visual Stream**: RGB-D (has different resolution), optional fisheye/multi-view.

**End-Effector Perception**: End-effector pose, joint angles, gripper status.

**Instruction Text**: Original prompt + 20 crowdsourced/rewrite variants.

**Calibration Information**: Camera intrinsic and extrinsic parameters, timestamp synchronization.

**Quality Labels**: Success/failure, abnormal frame markings, occlusion/shift indicators.

To align with the NoT-VLA intermediate interface, the data is serialized in JSON + NPZ (or Parquet): `meta.json` stores instructions, camera parameters, and time references; `traj.npz` stores image indices, depth, UV keypoints, discretized end-effector pose, gripper flag, and auxiliary quantities for the detokenizer. All entries are hash-checked and versioned; raw video frames are optionally packaged in H.264 with keyframe indexing for fast replay.

## C    DEPTH SENSING WITH ANCHOR

Acquiring accurate, reliable, and task-aligned depth data is crucial for manipulation tasks in 3D space. To obtain such depth data, the following approaches based on depth information can be adopted:

1. Direct sensor depth (applicable to both simulated and real-world scenarios);

2. Output from external monocular depth models;

3. Depth anchor tokens predicted by NoTVLA itself on the image plane, followed by relative depth calculation using prior object dimensions.

This "anchor-relative depth" paradigm significantly narrows the search space of the Vision-Language Model (VLM), enhancing generalization robustness across different views/scenes while preserving the *plug-and-play capability* of sensors and models.

### C.1    TRAINING STEPS

As shown in Table 7, the result presents the performance metrics (presumably success rates, given their range of 0 to 1) of 11 tasks categorized into Short Horizon, Medium Horizon, and Long Horizon groups, evaluated across 8 step horizons (500, 1000, 1500, 2000, 2500, 3000, 3500, and 4000 steps). The performance data across different step horizons reveals critical insights into the interplay between task complexity, horizon length, and system adaptability: First, task inherent difficulty overrides step horizon adjustments—Short Horizon Tasks, though theoretically less complex, show extreme disparities (e.g., "grab roller" and "place empty cup" consistently above 0.6, while "place phone stand" never exceeds 0.25), indicating that task-specific challenges (e.g., object shape, precision requirements) have a more significant impact on success than step count. Second, system stability correlates with task type rather than horizon length: Medium Horizon Tasks like "stack blocks two" maintain steady performance (0.50–0.75 across all steps) because stacking tasks likely have more predictable action sequences, whereas erratic results in "place shoe" (Short Horizon) and "place a2b random" (Medium Horizon) suggest the system struggles with tasks involving variable object positions or unstructured interactions. Third, long-horizon tasks benefit from moderate step increases but hit diminishing returns: "stack blocks three" and "blocks ranking rgb" improve from 500 to 3500 steps (peaking at 0.60 and 0.70 respectively) but do not continue to advance at 4000 steps, implying that beyond a certain threshold, additional steps do not compensate for the cumulative complexity of multi-step tasks (e.g., error propagation in block stacking).

Table 7: Task performance comparison at various horizons (Short, Medium, Long).

| Task Name | 500 | 1000 | 1500 | 2000 | 2500 | 3000 | 3500 | 4000 |
|---|---|---|---|---|---|---|---|---|
| **Short Horizon Tasks** | | | | | | | | |
| grab roller | 0.85 | 0.80 | 0.70 | 0.85 | 0.90 | 1.00 | 0.90 | 0.80 |
| place phone stand | 0.00 | 0.25 | 0.05 | 0.10 | 0.15 | 0.25 | 0.15 | 0.15 |
| place empty cup | 0.75 | 0.85 | 0.75 | 0.70 | 0.65 | 0.60 | 0.70 | 0.80 |
| place shoe | 0.10 | 0.25 | 0.30 | 0.55 | 0.30 | 0.15 | 0.50 | 0.25 |
| **Medium Horizon Tasks** | | | | | | | | |
| place a2b random | 0.05 | 0.20 | 0.40 | 0.25 | 0.35 | 0.50 | 0.25 | 0.20 |
| place bread basket | 0.25 | 0.50 | 0.60 | 0.45 | 0.40 | 0.40 | 0.55 | 0.40 |
| stack blocks two | 0.50 | 0.75 | 0.65 | 0.75 | 0.75 | 0.75 | 0.65 | 0.75 |
| stack colorfulblocks two | 0.55 | 0.65 | 0.50 | 0.65 | 0.55 | 0.65 | 0.40 | 0.70 |
| **Long Horizon Tasks** | | | | | | | | |
| stack blocks three | 0.05 | 0.30 | 0.25 | 0.45 | 0.50 | 0.30 | 0.60 | 0.50 |
| blocks ranking rgb | 0.25 | 0.45 | 0.60 | 0.55 | 0.50 | 0.45 | 0.70 | 0.50 |

## C.2 TRAJECTORY PLANNING

Table 9 compares the performance of NoTVLA planning trajectory in 2D and Magma planning trajectory in 2D across 10 key metrics, which evaluates the accuracy, similarity, and spatial alignment of generated or processed shapes. A critical distinction in metric interpretation first emerges: while higher values indicate better performance for cover F1 and cover precision (standard classification/segmentation metrics where 1.0 represents perfect performance), lower values signal superior results for the remaining eight metrics (e.g., DTW, endpoint error, Frechet distance), as these quantify discrepancies or distances between target and predicted shapes—smaller values mean less deviation from the ground truth. Against this framework, NoTVLA planning trajectory in 2D outperforms Magma planning trajectory in 2D across all metrics, demonstrating consistent superiority in both shape coverage and spatial accuracy. For the coverage-related metrics, NoTVLA planning trajectory in 2D achieves a cover F1 score of 0.9546, which is approximately 2.8% higher than Magma Shape's 0.9282, and a cover precision of 0.9558 (nearly 0.8% higher than Magma Shape's 0.9484); these gains indicate that NoTVLA planning trajectory in 2D not only captures more of the target shape (better coverage) but also does so with fewer false positives (higher precision). For DTW (Dynamic Time Warping, a measure of temporal or sequential similarity), NoTVLA planning trajectory in 2D scores 0.4565—less than half of Magma Shape's 0.8353—suggesting that if the shapes are time-dependent (e.g., motion trajectories), "Our Shape" aligns far more closely with the target sequence. Similarly, NoTVLA planning trajectory in 2D reduces endpoint error (a measure of how much the final point of the shape deviates from the target) by over 58% (0.0740 vs. 0.1785) and Frechet distance (a metric for comparing continuous shapes) by nearly 55% (0.1035 vs. 0.2297), both of which are critical for tasks where precise shape boundaries or endpoints matter (e.g., robotic path planning or medical image segmentation). Even in finer-grained spatial metrics—such as mean orthogonal distance (average deviation across all points) and median orthogonal distance (middle value of deviations, less sensitive to outliers)—NoTVLA planning trajectory in 2D maintains a clear edge: its mean orthogonal distance (0.0322) is roughly 32% lower than Magma Shape's (0.0476), and its median (0.0286) is 35% lower, confirming that this superiority is consistent across most points of the shape.

## C.3 ZERO SHOT OF DIFFERENT MODELS

The Table 8 evaluates the zero-shot performance of three NoTVLA models (differentiated by their parameter sizes: 7B, 3B, and 1B) across three distinct task categories—Color Recognition, Target Selection, and Spatial Semantics—offering key insights into how model scale influences generalization ability in zero-shot settings, where models are tested on tasks without explicit prior training. A foundational observation from the data is the clear correlation between model parameter size and overall zero-shot performance: larger models consistently outperform smaller ones across nearly all tasks, while the smallest 1B parameter model shows minimal to no success in most cases, highlighting a critical threshold in parameter scale for enabling effective zero-shot generalization.

Table 8: ZEROSHOT task performance comparison across different models.

| Task Name | NoTVLA 7B | NoTVLA 3B | NoTVLA 1B |
|---|---|---|---|
| **Color Recognition** | | | |
| stack block two(easy) | 0.86 | 0.84 | 0.00 |
| stack random color blocks two(easy) | 0.69 | 0.55 | 0.00 |
| **Target Selection** | | | |
| move pillbottle pad(hard) | 0.38 | 0.21 | 0.00 |
| move pillbottle pad(easy) | 0.38 | 0.27 | 0.00 |
| move pillbottle pad choice(hard) | 0.26 | 0.14 | 0.00 |
| move pillbottle pad choice(easy) | 0.28 | 0.20 | 0.00 |
| **Spatial Semantics** | | | |
| place a2b left(easy) | 0.39 | 0.21 | 0.04 |
| place a2b right(easy) | 0.28 | 0.17 | 0.00 |
| place a2b randomly(hard) | 0.36 | 0.19 | 0.02 |
| place a2b randomly(easy) | 0.34 | 0.24 | 0.02 |

Table 9: Comparison of performance metrics across different methods.

| Metric | Our Shape | Magma Shape |
|---|---|---|
| cover f1 | 0.954600 | 0.928213 |
| cover precision | 0.955800 | 0.948400 |
| dtw | 0.456478 | 0.835321 |
| endpoint err | 0.073954 | 0.178488 |
| frechet | 0.103547 | 0.229654 |
| hausdorff | 0.080929 | 0.128247 |
| max orth dist | 0.074424 | 0.110651 |
| mean orth dist | 0.032231 | 0.047644 |
| median orth dist | 0.028614 | 0.043905 |
| startpoint err | 0.077855 | 0.144978 |

In the Color Recognition category, which focuses on tasks requiring identification and manipulation of colored blocks, the performance gap between model sizes is particularly striking. The 7B model maintains strong effectiveness here, while the 3B model exhibits a modest drop in performance relative to the 7B variant—suggesting that color-related zero-shot tasks demand a certain level of model capacity to encode and apply visual-semantic associations, but that even the 3B model retains some ability to generalize. In contrast, the 1B model fails entirely at these tasks, indicating that its capacity is insufficient to capture the basic visual cues or task logic needed for recognizing and stacking colored blocks without prior training. The Target Selection category, which involves tasks centered on manipulating specific objects (pillbottle pads) and includes both "easy" and "hard" difficulty levels, reinforces the size-performance trend while also revealing how task complexity interacts with model capacity. Across all Target Selection tasks, the 7B model achieves the highest success rates, with the 3B model showing noticeable underperformance—especially in the harder variants of the task. This suggests that as task difficulty increases (e.g., when choices or precision requirements are added), the gap between larger and smaller models widens, as larger models have more capacity to handle the additional complexity of decision-making or fine-grained object interaction in zero-shot scenarios. The 1B model's complete failure here further underscores that even relatively straightforward target-focused tasks require more than 1B parameters to generalize to without training.

In the Spatial Semantics category, which tests the ability to understand and act on spatial relationships (e.g., placing objects to the left or right, or in random positions), the same pattern holds, but with a subtle distinction: the 1B model shows minimal success in a small subset of tasks (e.g., "place a2b left (easy)") rather than complete failure. This tiny improvement, while negligible in practical terms, hints that spatial semantics tasks may have some simpler sub-components that the 1B model can partially grasp—though its overall inability to perform consistently across even easy spatial tasks confirms its limited capacity. For the 7B and 3B models, spatial tasks follow the familiar trend: the 7B model outperforms the 3B model, with both showing better performance on easier variants of the tasks, emphasizing that spatial reasoning in zero-shot settings, like color recognition and target selection, relies heavily on model scale to encode and apply abstract spatial concepts.

Collectively, these results point to two core conclusions: first, model parameter size is a primary determinant of zero-shot performance for the NoTVLA series, with a sharp drop-off in effectiveness when scaling down from 3B to 1B parameters—indicating that 3B parameters may represent a minimum threshold for meaningful zero-shot generalization across the tested task types. Second, task complexity (as reflected in "easy" vs. "hard" labels) interacts with model size: harder tasks amplify the performance gap between larger and smaller models, as they require more capacity to handle uncertainty, decision-making, or fine-grained reasoning. This suggests that for zero-shot applications involving visual manipulation or spatial reasoning, investing in larger model sizes (e.g., 7B parameters) is critical for achieving reliable performance, while smaller models (especially 1B) are unlikely to be viable unless tasks are extremely simple or include explicit training data.

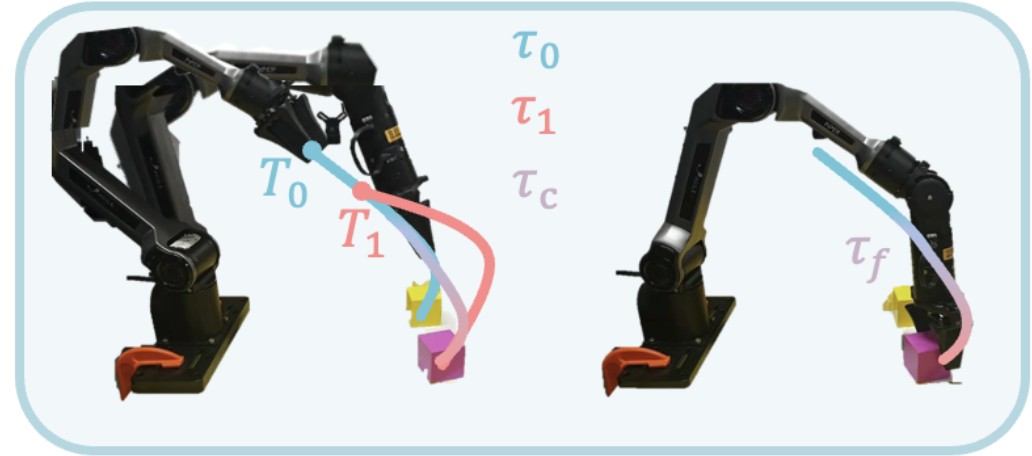

Figure 5: The close loop operation of NoTVLA in practice. $T_0$ was the first time NoTVLA infered and $T_1$ was the second time. But when the result of $T_1$ came out, which was $\tau_1$, the trajectory has followed to $\tau_0$. Then we need to combine $\tau_0$ and $\tau_1$ for a trajectory $\tau_c$. With $\tau_c$ combined step by step, we finally get the trajectory $\tau_f$.

## D  CROSS EMBODIMENT

A RealSense D455 depth camera is used, and camera calibration is performed via automatic calibration supplemented with manual fine-tuning to ensure the highest possible accuracy of camera extrinsic parameters.

Considering the incomparability of joint spaces across different robots, *normalized EE poses* are adopted as the unified control interface; joint angles are only used for diagnosis and visualization. Unified gripper coordinate system mapping and workspace constraints are established for six robotic platforms: Aloha-AgileX, ARX-X5, Franka Panda, UR5, Piper, and AgiBot G1. For cross-platform deployment, only the Action Expert (e.g., Inverse Kinematics (IK), Diffusion Policy, ACT, etc.) needs to be replaced to generate low-level control signals from the unified EE trajectory.

### D.1  CLOSE-LOOP

Achieving closed-loop control is a foundational goal for both Vision-Action (VA) and VLA models, as it ensures control stability in the presence of external disturbances (e.g., slight shifts in object position, minor robotic drift) or unmodeled dynamics (e.g., variations in friction between the end effector and a manipulated object). To realize this, our spline-based action detokenizer integrates a mechanism for real-time secondary inference, designed to enable trajectory correction without interrupting ongoing operation. The workflow operates as follows: upon initiating execution of a trajectory generated via the initial model inference, the VLA system simultaneously triggers a secondary inference process. This secondary inference leverages the latest visual feedback (e.g., a camera frame captured mid-execution) to re-estimate the target trajectory, accounting for any deviations from the original plan that may have occurred since execution began. During the interval when secondary inference is in progress, the robot continues to execute the original trajectory—this avoids motion halts and maintains task continuity, which is particularly critical for time-sensitive operations.

Once the secondary inference is completed, the system performs trajectory merging to integrate the updated plan into the ongoing execution. Rather than abruptly switching to the new trajectory (which would introduce discontinuities in velocity or acceleration), the detokenizer generates a short, spline-interpolated transition segment that connects the end effector's current position and velocity to the starting point of the newly inferred trajectory. This transition segment ensures smooth

motion throughout the handoff, while the remaining portion of the original trajectory is discarded to prevent further accumulation of errors. By iterating this process at fixed intervals (e.g., every 0.5 seconds), the system achieves continuous closed-loop control: each secondary inference cycle corrects for recent deviations, and the spline-based merging ensures that these corrections do not compromise trajectory smoothness. For example, if a target object shifts 2 cm horizontally during the "pre-grasp" phase of a task, the secondary inference will adjust the "post-grasp" trajectory to align with the object's new position, and the transition segment will ensure the end effector adapts to this shift without jerk or delay.

Take the real situation as an example. At $t_0$, waypoints $\mathcal{P}_0 = \{(u_i, v_i, d_i)\}_{i=1}^n$ are projected to 3D using intrinsics $\boldsymbol{K}$ via $\boldsymbol{p}_i = d_i \boldsymbol{K}^{-1}[u_i, v_i, 1]^\top$, then fit with a cubic B-spline (centripetal parameterization) to produce $\boldsymbol{T}_0(t)$ ($C^2$ continuous inside segments). At $t_1$, new waypoints $\mathcal{P}_1$ arrive. At $t_2 = t_1 + \Delta t$, we update from the current pose $\boldsymbol{T}_0(t_2)$, the executed subset of $\mathcal{P}_0$, and $\mathcal{P}_1$. We re-align the remaining (unexecuted) waypoint list by locating the closest still-pending waypoint to the current pose instead of using a fixed distance culling radius. Let the ordered (future) waypoint set be $\mathcal{P}' = \{(u_i', v_i', d_i')\}_{i=1}^{n'}$ with 3D back-projections $\boldsymbol{p}_i' = d_i' \boldsymbol{K}^{-1}[u_i', v_i', 1]^\top$. Define

$$k^* = \arg \min_{1 \leq i \leq n'} \left\| \boldsymbol{p}_i' - \boldsymbol{T}_0(t_2) \right\|_2. \tag{13}$$

We compute a local forward direction (preferred motion direction) as

$$\hat{\boldsymbol{v}}_{\text{path}} = \begin{cases} \dfrac{\boldsymbol{p}_{k^*+1}' - \boldsymbol{p}_{k^*}'}{\|\boldsymbol{p}_{k^*+1}' - \boldsymbol{p}_{k^*}'\|_2}, & k^* < n', \\ \dfrac{\boldsymbol{p}_{k^*}' - \boldsymbol{p}_{k^*-1}'}{\|\boldsymbol{p}_{k^*}' - \boldsymbol{p}_{k^*-1}'\|_2}, & k^* = n'. \end{cases} \tag{14}$$

We decide whether to keep the closest waypoint itself by a directional consistency test:

$$\gamma = (\boldsymbol{p}_{k^*}' - \boldsymbol{T}_0(t_2))^\top \hat{\boldsymbol{v}}_{\text{path}}, \qquad \text{keep } k^* \text{ iff } \gamma > 0. \tag{15}$$

All earlier indices are discarded. Let the drop indicator be $\delta_{k^*} = \mathbb{1}[\gamma \leq 0]$. The refreshed pending waypoint sequence is then

$$\mathcal{P}_{\text{pending}} = \{(u_j', v_j', d_j') \mid j \geq k^* + \delta_{k^*}\}. \tag{16}$$

# E  MORE RESULTS AND REALITY PERFORMANCE

## E.1  ROBOTWIN 2.0 FULL RESULTS

The performance comparison of ACT, DP, DP3 $\pi_0$, RDT and NoTVLA is shown in Table 10.

Table 10: Performance comparison between expert models and generalist models on various robotic tasks with old loss data.

| Task Name | Expert Models | | | Generalist Models | | |
|---|---|---|---|---|---|---|
| | ACT | DP | DP3 | Pi0 | RDT | NoTVLA |
| click alarmclock | 0.32 | 0.61 | 0.77 | 0.63 | 0.55 | **0.78** |
| click bell | 0.58 | 0.54 | 0.90 | 0.44 | 0.92 | **0.94** |
| grab roller | 0.94 | **0.98** | **0.98** | 0.96 | 0.94 | 0.97 |
| handover mic | 0.85 | 0.53 | **1.00** | 0.98 | 0.97 | 0.99 |
| move playingcard away | 0.36 | 0.47 | 0.68 | 0.53 | 0.60 | 0.77 |
| open laptop | 0.56 | 0.49 | 0.82 | **0.85** | 0.62 | 0.57 |
| press stapler | 0.31 | 0.06 | 0.69 | 0.62 | 0.88 | **0.94** |
| shake bottle | 0.74 | 0.65 | **0.98** | 0.97 | 0.87 | 0.94 |
| Beat Block Hammer | 0.56 | 0.42 | 0.72 | 0.43 | 0.77 | **0.98** |
| Blocks Ranking RGB | 0.01 | 0.00 | 0.03 | 0.19 | 0.03 | **0.76** |
| Blocks Ranking Size | 0.00 | 0.01 | 0.02 | 0.07 | 0.00 | **0.53** |
| Click Alarmclock | 0.32 | 0.61 | 0.77 | 0.63 | 0.61 | **0.78** |
| Click Bell | 0.58 | 0.54 | 0.90 | 0.44 | 0.80 | **0.94** |
| Grab Roller | 0.94 | **0.98** | **0.98** | 0.96 | 0.74 | 0.97 |
| Handover Mic | 0.85 | 0.53 | **1.00** | 0.98 | 0.90 | 0.99 |
| Lift Pot | 0.88 | 0.39 | **0.97** | 0.84 | 0.72 | 0.90 |
| Move Can Pot | 0.22 | 0.39 | 0.70 | 0.58 | 0.25 | **0.76** |
| Move Pillbottle Pad | 0.00 | 0.01 | **0.41** | 0.21 | 0.08 | 0.12 |
| Move Playingcard Away | 0.36 | 0.47 | 0.68 | 0.53 | 0.43 | **0.77** |
| Move Stapler Pad | 0.00 | 0.01 | 0.12 | 0.00 | 0.02 | **0.28** |
| Open Laptop | 0.56 | 0.49 | 0.82 | **0.85** | 0.59 | 0.57 |
| Place A2B Left | 0.01 | 0.02 | **0.46** | 0.31 | 0.03 | 0.38 |
| Place A2B Right | 0.00 | 0.13 | 0.49 | **0.27** | 0.01 | 0.34 |
| Place Bread Basket | 0.06 | 0.14 | 0.26 | 0.17 | 0.10 | **0.63** |
| Place Burger Fries | 0.49 | 0.72 | 0.72 | **0.80** | 0.50 | 0.69 |
| Place Can Basket | 0.01 | 0.18 | 0.67 | 0.41 | 0.19 | **0.77** |
| Place Cans Plasticbox | 0.16 | 0.40 | 0.48 | 0.34 | 0.06 | **0.51** |
| Place Container Plate | 0.72 | 0.41 | 0.86 | 0.88 | 0.78 | **0.92** |
| Place Dual Shoes | 0.09 | 0.08 | 0.13 | 0.15 | 0.04 | **0.21** |
| Place Empty Cup | 0.61 | 0.37 | 0.65 | 0.37 | 0.56 | **0.90** |
| Place Fan | 0.01 | 0.03 | **0.36** | 0.20 | 0.12 | 0.30 |
| Place Mouse Pad | 0.00 | 0.00 | 0.04 | 0.07 | 0.01 | **0.23** |
| Place Object Scale | 0.00 | 0.01 | 0.15 | 0.10 | 0.01 | **0.43** |
| Place Object Stand | 0.01 | 0.22 | 0.60 | 0.36 | 0.15 | **0.75** |
| Place Phone Stand | 0.02 | 0.13 | **0.44** | 0.35 | 0.15 | 0.25 |
| Place Shoe | 0.05 | 0.23 | 0.58 | 0.28 | 0.35 | **0.59** |
| Press Stapler | 0.31 | 0.06 | 0.69 | 0.62 | 0.41 | **0.94** |
| Put Object Cabinet | 0.15 | 0.42 | 0.72 | 0.68 | 0.33 | **0.81** |
| Rotate QRcode | 0.01 | 0.13 | **0.74** | 0.68 | 0.50 | 0.73 |
| Scan Object | 0.02 | 0.09 | **0.31** | 0.18 | 0.04 | 0.24 |
| Shake Bottle | 0.74 | 0.65 | **0.98** | 0.97 | 0.74 | 0.94 |
| Shake Bottle Horizontally | 0.63 | 0.59 | **1.00** | 0.99 | 0.84 | 0.96 |
| Stack Blocks Three | 0.00 | 0.00 | 0.01 | 0.17 | 0.02 | **0.49** |
| Stack Blocks Two | 0.25 | 0.07 | 0.24 | 0.42 | 0.21 | **0.88** |
| Stack Bowls Three | 0.48 | 0.63 | 0.57 | 0.66 | 0.51 | **0.83** |
| Stack Bowls Two | 0.82 | 0.61 | 0.83 | 0.91 | 0.76 | **0.96** |
| Stamp Seal | 0.02 | 0.02 | 0.18 | 0.03 | 0.01 | **0.36** |

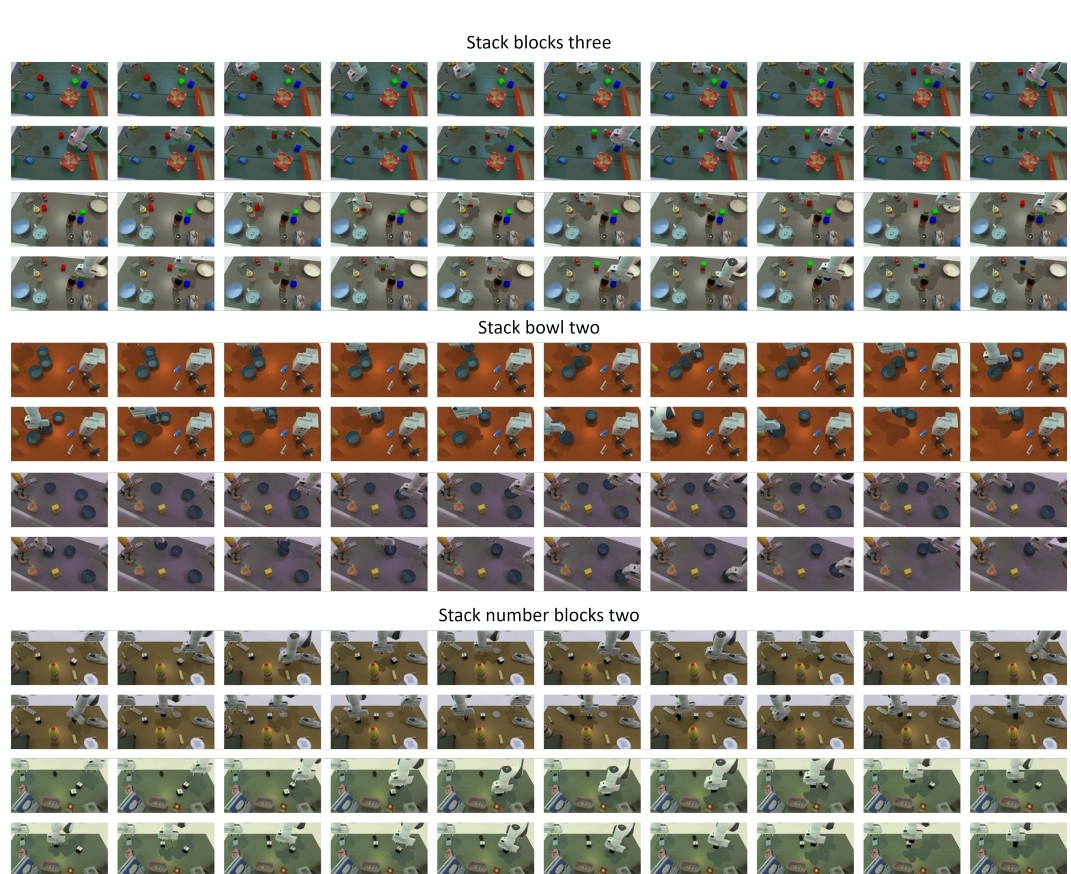

Figure 6: Franka Operation in the RoboTwin Simulation

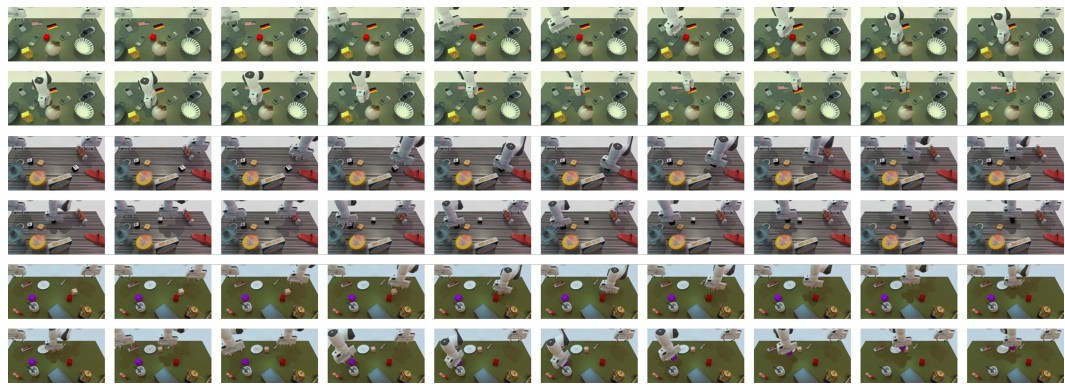

Figure 7: Zero Shot Franka Operation in the RoboTwin Simulation

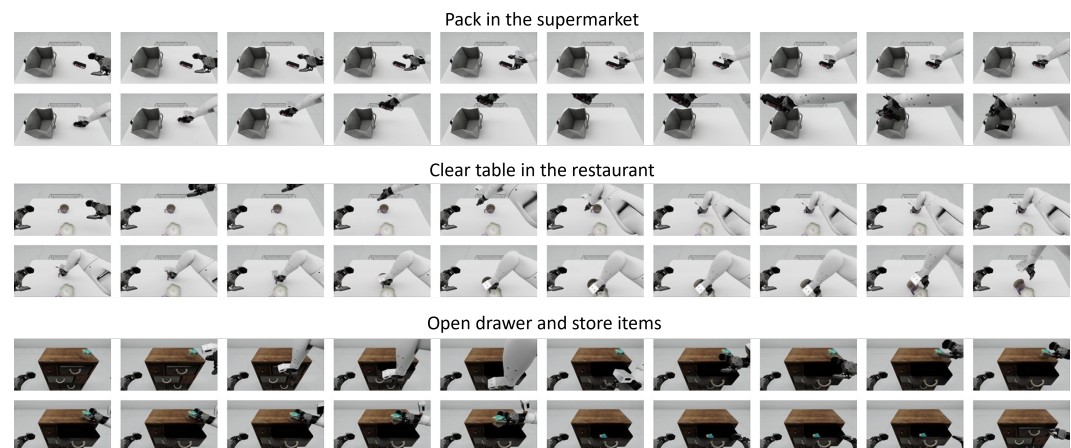

Figure 8: AGIBOT G1 Operation in the Simulation

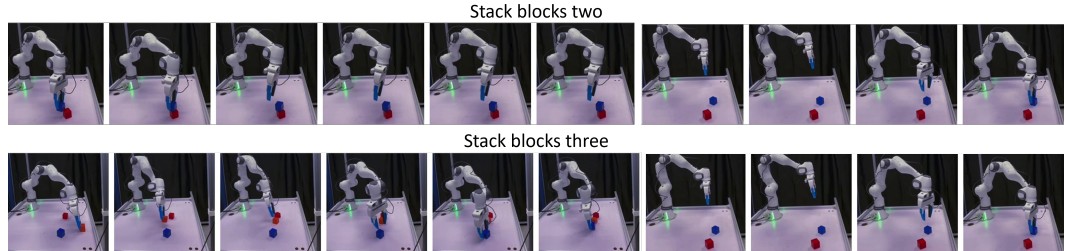

Figure 9: Franka Operation in the Reality

Table 11: Real-Robot Success Rates (SR). We test different task on PiPER and Franka platform.

| Robot | Task | SR |
|---|---|---|
| **PiPER** | Press button | 9/10 |
| | Place cube flag | 6/10 |
| | Stack cube two | 2/10 |
| **Franka** | Stack cube two | 5/10 |

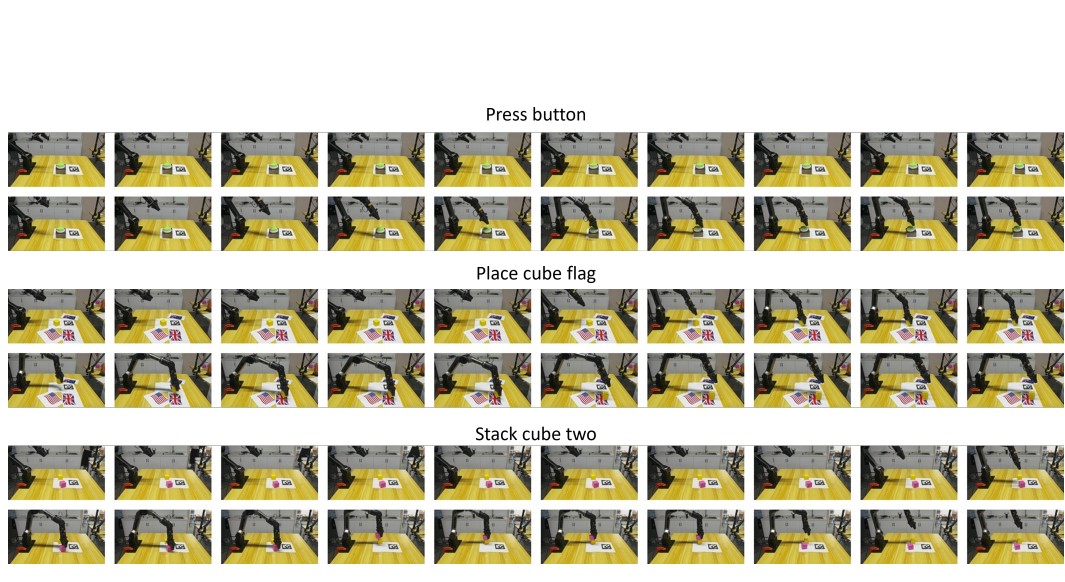

Figure 10: PiPER Operation in the Reality

