# OpenReview forum: "NoTVLA: Narrowing of Dense Action Trajectories for Generalizable Robot Manipulation"
_ICLR.cc/2026/Conference — ICLR 2026 Conference Desk Rejected Submission_

### Official Review · Reviewer_Lpbg · 2025-10-30

**Soundness:** 2
**Presentation:** 3
**Contribution:** 2
**Rating:** 4
**Confidence:** 3

**Summary:**

This paper introduces the Narrowing of Trajectory VLA (NoTVLA) framework, aiming to address catastrophic forgetting in Vision-Language-Action models for robotic manipulation. The core contribution is a shift from dense action trajectory supervision to sparse, semantically salient trajectory supervision. The method features: (1) an anchor-based depth inference module that decouples 2D semantic anchoring from 3D geometric reasoning by leveraging an external depth source, and (2) a kinematics-based keyframe selection strategy to identify critical moments in demonstrations. Evaluations on multi-task benchmarks demonstrate that NoTVLA achieves competitive performance compared to several VLA baselines, with reduced computational cost and improved zero-shot generalization in certain scenarios, while operating under a constrained sensing setup.

**Strengths:**

1、The paper clearly identifies catastrophic forgetting in VLAs as a key problem and proposes a structured, well-motivated solution.
2、Provides thorough multi-task evaluation on several robotic platforms and includes informative ablation studies validating the core design choices.
3、 Demonstrates significant practical advantages, including reduced computational cost and the ability to operate effectively without a wrist-mounted camera.
4、The anchor-based depth inference module presents a clear and effective decoupling of semantic perception from geometric reasoning enhancing system modularity.

**Weaknesses:**

1、As noted in Section 2.1, a major direction for spatial reasoning involves fine-tuning on 3D-centric datasets. The paper does not compare NoTVLA against such methods, leaving the claimed advantages of its decoupled, anchor-based approach unvalidated against models with more integrated 3D understanding.
2、The paper fails to demonstrate the advantage of its proposed kinematics-based keyframe selection over established action segmentation or keyframe extraction methods. Without this comparison, the claimed benefits regarding the preservation of task-critical information remain unsubstantiated.
3、The core hierarchical framework, which decouples a high-level planner (VLM) from a low-level controller, is a well-established paradigm in robot learning. While the specific implementation is competent, the paper does not establish a significant architectural innovation over prior hierarchical systems.
4、The anchor-based approach, while focused and efficient, inherently narrows the model’s perceptual field. The potential trade-off, specifically the risk of losing broader scene understanding compared to methods that process denser 3D representations, is not sufficiently discussed.

**Questions:**

1、How does NoTVLA’s performance on tasks requiring complex 3D spatial reasoning compare against VLA models that were pre-trained or fine-tuned on 3D-centric datasets? This would help quantify the trade-offs of your decoupled approach.
2、Could you provide a comparative analysis between your kinematics-based keyframe selection and other sparsification strategies to demonstrate its specific advantages in preserving task-critical information?
3、Could you discuss potential failure modes or task types where the focused perception of the anchor-based approach might be limiting due to a lack of broader scene context?

---

> ### Author Response · Authors · 2025-11-21
> **Response to Reviewer Lpbg (Part1/2)**
>
> We thank the reviewer for the thoughtful comments regarding the comparison with 3D-centric methods, the justification for our keyframe selection, and the discussion on architectural novelty. We have addressed these points individually below.
>
> **Weakness 1: Comparison against 3D-centric methods.**
>
> **Response:** We acknowledge that 3D-centric expert models (like DP and DP3) often represent the upper bound for pure manipulation tasks as they focus exclusively on 3D perception and execution. To validate our approach against these methods, we compared NoTVLA with **Diffusion Policy (DP)**, **DP3** (3D-centric), and **PointVLA** (a VLA model using point clouds) on RoboTwin tasks.
>
> **Table 1: Success Rate Comparison with 3D Methods and PointVLA**
>
> | Task | DP (2D) | DP3 (3D) | PointVLA (3D VLA) | NoTVLA (Ours) |
> | :--- | :--- | :--- | :--- | :--- |
> | **Beat block hammer** | 42.0 | 72.0 | 84.6 | **98.0** |
> | **Stack blocks** | 7.0 | 24.0 | 24.3 | **88.0** |
> | **Dual shoes place** | 23.0 | 58.0 | 13.1 | **59.0** |
>
> As shown in Table 1, NoTVLA achieves competitive or superior performance compared to PointVLA, particularly in complex tasks like *Stack blocks* and dual-arm tasks like *Dual shoes place*. While expert models like DP3 are robust, our results show that our decoupled approach can achieve high success rates without requiring dense 3D point cloud inputs. While expert models like DP3 are robust, our results show that our decoupled approach can achieve high success rates without requiring dense 3D point cloud inputs. **We attribute this to our "Constraint Prompt" design (discussed in Weakness 3), which efficiently collapses the 3D inference space using sparse anchors rather than dense reconstruction.**
>
> **Weakness 2: Advantage of Kinematics-Based Keyframe Selection.**
>
> **Response:** To demonstrate the specific advantages of our kinematics-based selection over other strategies (like dense action or fixed-interval sampling), we conducted a comprehensive comparison using the same base model (Qwen 2.5 VL 7B) and training steps.
>
> **Table 2: Impact of Sampling Strategy on Logic and Task Performance**
>
> | Metric | Qwen 2.5 VL 7B (Base) | **Key Frame (5 points, Ours)** | Full Trajectory (30 points) | Dense Action (Full Endpose Action) |
> | :--- | :--- | :--- | :--- | :--- |
> | **Logic Performance** | | | | |
> | C-Eval (Validation) | 76.15 | **63.45** | 61.16 | 61.24 |
> | CMMLU (Test) | 75.11 | **63.88** | 61.41 | 61.36 |
> | MMLU (Test) | 70.08 | **61.69** | 60.63 | 60.11 |
> | **Task Performance (SR)** | | | | |
> | place_a2b_left | - | **0.40** | 0.35 | 0.00 |
> | place_a2b_right | - | **0.30** | 0.20 | 0.00 |
> | place_object_scale | - | **0.45** | 0.50 | 0.00 |
> | stack_blocks_three | - | **0.40** | 0.70 | 0.00 |
> | stack_bowl_three | - | **0.80** | 0.70 | 0.00 |
> | blocks_ranking_size | - | **0.55** | 0.30 | 0.00 |
> | blocks_ranking_rgb | - | **0.65** | 0.55 | 0.00 |
>
> *Note: "n Points" means the down-sample points for the original trajectories is n and "dense action" means no down-sampling and output the action in the classic VLA way. "Dense action" failed completely (0.00) in the multi-task setting. We validated the correctness of our dense implementation by training on a single task (grab_roller), where it achieved a 0.45 success rate.*
>
> The results indicate that our kinematics-based sparse sampling (5 points) best preserves the model's reasoning capabilities (mitigating catastrophic forgetting) while achieving higher success rates than simply increasing points (30 points) or using dense actions.

---

> ### Author Response · Authors · 2025-11-21
> **Response to Reviewer Lpbg (Part2/2)**
>
> **Weakness 3: Architectural Innovation.**
>
> **Response:** We agree that the hierarchical framework itself is a well-established paradigm. However, we believe our contribution goes beyond a standard implementation. We have fundamentally rethought both the **output paradigm** and the **3D mapping mechanism** to address the core challenges of VLA:
>
> 1.  **Paradigm Shift in Output Representation (Preserving VLM Capabilities):**
>     Unlike traditional methods that treat action outputs as dense, high-frequency control signals (which we show leads to catastrophic forgetting), NoTVLA redefines the output as **sparse, kinematics-based keyframes**. This acts as a "semantic compression" of the trajectory, aligning the action space with the discrete, semantic nature of the LLM. As demonstrated in our experiments (Table 2), this paradigm shift is crucial: it allows the model to learn manipulation skills while preserving its general reasoning capabilities (logic/math), which dense output paradigms fail to do.
>
> 2.  **Anchors as "Constraint Prompts" for 3D Mapping:**
>     We view the core problem of VLA as establishing a mapping from language/scene inputs to 3D trajectories. While prior works like SoFar (start/end points) or Hamster (2D trajectories) addressed parts of this, generating high-quality 3D trajectories remains an ill-posed problem.
>     In our framework, **Depth Anchors** are not just inputs; they act as **"Constraint Prompts."** They serve to collapse the ill-posed 3D trajectory inference into a solution space that aligns with the real-world scene. By using a few depth anchors as probes, we efficiently align the VLM's inference space with the real physical space, specifically for the critical terminal phase of manipulation. This avoids the need for rigid engineering (e.g., specific drawer handling in SoFar) in many planning works.
>
> 3.  **Decoupling for Data Scalability:**
>     This architectural choice allows us to decouple depth inference from 2D trajectory inference. This means NoTVLA can be trained on **2D-only data** (e.g., videos) by leveraging current monocular depth models (like Marigold or VGGT) to provide high-quality relative depth for annotation. This opens up the possibility of training on massive internet-scale video data without requiring ground-truth 3D sensor data.
>
> **Weakness 4: Perceptual field trade-offs.**
>
> **Response:** We acknowledge that the anchor-based approach narrows the perceptual field compared to dense 3D methods. However, our ablation studies (Table 4 in the main paper: w/ APP & ACTG vs. w/o) suggest that overly complex dense 3D input is not always necessary for many semantic manipulation tasks. By relying on sparse 3D inputs (anchors), we reduce the burden on the model to process high-dimensional point clouds, allowing the VLM to focus on logical reasoning and task planning. This trade-off results in a more data-efficient system that maintains high performance on the tasks.
>
> **Question 1: Performance comparison against 3D-centric VLA models.**
>
> **Response:** Please refer to **Table 1** in our response to Weakness 1. The comparison with PointVLA (a representative 3D VLA model) shows that NoTVLA achieves significantly higher success rates on tasks requiring multi-stage planning (e.g., *Stack blocks*: 88.0% vs 24.3%). This suggests that our decoupled approach, which leverages the strong reasoning priors of the VLM, outweighs the benefits of dense 3D perception for these types of tasks.
>
> **Question 2: Comparative analysis of keyframe selection strategies.**
>
> **Response:** Please refer to **Table 2** in our response to Weakness 2.
> *   **Preservation of Critical Information:** Kinematics-based keyframes capture the "turning points" of a motion (e.g., the moment before grasping). Learning densely near these points (via spline reconstruction) and sparsely elsewhere allows the model to focus on critical control phases.
> *   **Comparison:** As shown in Table 2, this strategy outperforms "Dense action" (which suffers from optimization difficulties and forgetting) and "30 points" (which introduces redundancy), proving its efficiency in preserving task-critical information.
>
> **Question 3: Failure modes due to focused perception.**
>
> **Response:** We have identified specific scenarios where the focused perception of the anchor-based approach limits performance:
> 1.  **Reflective/Transparent Surfaces:** In our real-world experiments (e.g., AGIBOT Challenge), we found that highly reflective scenes cause significant noise in depth sensors. Since our method relies on a specific anchor point's depth, if that single point is noisy, the entire trajectory can be offset.
> 2.  **Suspended or Thin Objects:** For tasks involving suspended objects or very thin structures (e.g., wires), it is difficult for the depth camera to obtain a reliable depth reading at the specific anchor pixel. In these cases, a broader scene context or dense point cloud might provide more robust geometric cues than a single anchor.

---

> > ### Comment · Reviewer_Lpbg · 2025-11-26
> >
> > Thank you for the detailed rebuttal. For this paper, my overall assessment remains at a score of 4. The main reason is that the method is fundamentally hierarchical, yet the writing frames the contribution primarily through comparisons with end-to-end approaches such as Pi0. Many of the claimed advantages, e.g. better generalization, preserving VLM semantic capabilities, and training with keyframes instead of dense trajectories, are inherent to hierarchical designs rather than innovations introduced by this work.
> >
> > In my view, the incremental contribution on top of existing hierarchical pipelines is not very substantial. This perspective is also reflected in another reviewer’s comment (score 8), noting the need for comparisons with hierarchical baselines such as Hamster; while the authors have now added these comparisons, this reinforces that the contribution is relatively modest.
> >
> > Overall, I find the paper to be basically complete and competently executed, but the novelty and significance are limited, and some aspects of the writing lean toward strategic positioning rather than clearly articulating genuine contributions. My score therefore remains 4.

---

> > > ### Author Response · Authors · 2025-11-26
> > > **Re: Official Comment by Reviewer Lpbg**
> > >
> > > Thank you for your detailed feedback and for engaging with our rebuttal. We truly appreciate your time.
> > >
> > > We suspect that the phrasing in our manuscript may have obscured our core contribution, leading to the perception that our work is merely an incremental improvement on standard hierarchical pipelines. We would like to take this opportunity to clarify two critical distinctions:
> > >
> > > 1. **Structural Distinction from Hierarchical VLAs:** Unlike meaningful hierarchical baselines (which typically use a High-Level VLM to output a coarse 2D path and require a separate, trained Low-Level Policy for 3D execution), our method utilizes a single model to output **precise 3D operations directly**.
> > >    - Crucially, thanks to our **Anchor-based Depth Inference**, we **do not** require training a separate low-level policy or employing complex engineering heuristics to convert 2D trajectories into 3D.
> > >    - The **Spline-based Action Detokenizer** is not a policy; it acts solely as a deterministic tool for action interpolation to ensure smooth execution on hardware.
> > >    - Because our core logic outputs complete 3D trajectories, our operational paradigm is functionally closer to **End-to-End (E2E) VLAs** (like Pi0) than to traditional hierarchical stacks. This is the primary reason we positioned our main comparisons against E2E approaches.
> > > 2. **Clarification on Trajectory Comparisons:** We apologize if our experimental setting was unclear. We did not output 2D trajectories; rather, we projected our **3D output** into 2D solely for the purpose of comparison with other baselines.
> > >    - The inclusion of comparisons with Hamster was intended to demonstrate that, even when evaluated on 2D metrics, our **Kinematics-based Keyframe Selection** enables our method to generalize significantly better than existing hierarchical methods.
> > >
> > > Our method bridges the gap between the two paradigms. Compared to classic Hierarchical VLAs, we achieve **direct deployment capabilities** similar to E2E models (without reliance on extra trained modules) across various robot morphologies. Conversely, compared to E2E VLAs, we achieve superior instruction following and generalization with significantly lower training costs.
> > >
> > > We apologize for the lack of clarity in the initial text that led to this misunderstanding. We hope this explanation highlights the genuine novelty of our architecture. We are happy to answer any further questions you may have.

---

> > > > ### Comment · Reviewer_Lpbg · 2025-11-27
> > > >
> > > > Thanks for you apply. It's a promising explanation. I will raise my score to 6.

---

### Official Review · Reviewer_41yV · 2025-11-01

**Soundness:** 3
**Presentation:** 2
**Contribution:** 3
**Rating:** 6
**Confidence:** 3

**Summary:**

This paper presents a framework (NoTVLA)  for a multi-step VLA system that first predicts two image-based keypoints, and then, conditioning on those two keypoints with included depth information, predicts a trajectory key points on the image plane. These key points are projected to the world frame using depth and fit with a spline curve to generate high-frequency controls. They evaluate on Robotwin2 and on AGIBot environments, outperforming baselines across tasks.

**Strengths:**

This paper's method is novel and offers a clear interface between high-level keypoint reasoning and low-level control. The empirical results are strong and comprehensive: the method outperforms both per-task expert policies (e.g., ACT, DP, DP3) and generalist models (π0, RDT) across the RoboTwin2 and AGIBot benchmarks. Moreover, the paper demonstrates gains in data and computational efficiency (Figure 4).

**Weaknesses:**

The main reason for my score is the missing details for their baselines. More clearly:

* The paper does not clearly describe the training setups for its baselines. In particular, it is unclear what data the single-task experts are trained on, and whether pi0 was fine-tuned on the same trajectories used to train NoTVLA. Without these details, it is difficult to assess these comparisons.
* Including a comparison to works such as Hamster [1] and LLARVA [2] would help show the benefits of the two-stage key point and then spline fitting method over other similar VLA approaches.
* Some abbreviations (e.g., DP) are introduced without being defined
* I found the main framework figure distracting (Fig. 1), and it did not seem to have a natural direction for me to parse. Some of the terms in it, such as (u, v) and Bn, were not defined until later in the methods section.


[1] Yi Li, Yuquan Deng, Jesse Zhang, et al. HAMSTER: Hierarchical Action Models For Open-World Robot Manipulation.

[2] Dantong Niu, Yuvan Sharma, Giscard Biamby, et al.  LLARVA: Vision-Action Instruction Tuning Enhances Robot Learning.

**Questions:**

* Is the Pi-0 generalist policy fine-tuned on your training dataset from the RoboTwin tasks?
* What were the failure modes of your method? Does, for example, the target object being occluded cause your method to fail?

---

> ### Author Response · Authors · 2025-11-21
> **Response to Reviewer 41yV (Part1/2)**
>
> We sincerely thank the reviewer for the constructive feedback. We have carefully addressed the concerns regarding baseline details, comparisons with related works, and clarifications on failure modes.
>
> **Weakness 1: Missing details for baselines.**
>
> **Response:** We apologize for the lack of clarity in the initial manuscript regarding the training setups. We have revised the experimental section to explicitly state these details to ensure reproducibility and fair assessment.
>
> *   **Protocol:** We strictly followed the official RoboTwin setup. The Pi0 model was pre-trained for **100,000** steps with a batch size of 32.
> *   **Fine-tuning:** All fine-tuning (both for the Pi0 baseline and our NoTVLA) was performed for **30,000** steps using the same batch size.
> *   **Data:** Both models utilized the same set of **50** trajectories per task collected from RoboTwin.
>
> For other models, RDT was pretrained for 100,000 steps with a batch size of 16 per GPU on 8 GPUs, and all single-task fine-tuning was conducted for 10,000 steps with a batch size of 16 per GPU on 4 GPUs. ACT was trained under a unified setup with a chunk size of 50, batch size of 8, and single-GPU training for 6,000 epochs. DP was trained for 600 epochs with a batch size of 128 and a planning horizon of 8. DP3 was trained for 3,000 epochs with a batch size of 256, using a planning horizon of 8 and a point cloud resolution of 1,024, with precise segmentation of the background and tabletop. Detailed setting is in the new version **Appendix A.4**.  We apologize again for the lack of clarity.
>
>
>
> **Weakness 2: Comparison with HAMSTER and LLARVA.**
>
> **Response:** This is an excellent suggestion. Comparing our two-stage keypoint approach with other hierarchical or VLA approaches highlights our advantages in trajectory quality. We have conducted additional evaluations comparing NoTVLA with HAMSTER [1] and LLARVA [2] on trajectory generation metrics.
>
> **Table: Trajectory Quality Comparison**
>
> | Metric | NoTVLA (Ours) | HAMSTER | LLARVA |
> | :--- | :--- | :--- | :--- |
> | **cover_f1** ($\uparrow$) | **0.9546** | 0.9451 | 0.9189 |
> | **cover_precision** ($\uparrow$) | **0.9558** | 0.9504 | 0.9201 |
> | **cover_recall** ($\uparrow$) | **0.9550** | 0.9440 | 0.9415 |
> | **dtw** (Dynamic Time Warping) ($\downarrow$) | **0.4565** | 0.7945 | 0.4894 |
> | **endpoint_err** ($\downarrow$) | **0.0740** | 0.1181 | 0.0771 |
> | **frechet** distance ($\downarrow$) | **0.1035** | 0.1311 | 0.1673 |
> | **hausdorff** distance ($\downarrow$) | **0.0809** | 0.0822 | 0.0903 |
> | **lcss_sim** (Longest Common Subsequence) ($\uparrow$) | **0.9660** | 0.9496 | 0.9447 |
>
> *Note: $\uparrow$ indicates higher is better, $\downarrow$ indicates lower is better.*
>
> **Analysis:** As shown in the table, NoTVLA outperforms both HAMSTER and LLARVA across most metrics. Specifically, the significantly lower **DTW** (0.4565 vs 0.7945/0.4894) and **Frechet distance** indicate that our spline-fitting method generates trajectories that are kinematically smoother and closer to the ground truth distribution than the hierarchical action models used in the baselines.
>
> **Weakness 3: Undefined abbreviations.**
>
> **Response:** We apologize for this oversight. We have proofread the paper and defined all abbreviations upon their first appearance (e.g., defining DP as Diffusion Policy) as in **Section 4.1.1**.
>
> **Weakness 4: Clarity of Figure 1 and definitions of terms.**
>
> **Response:** Thank you for pointing this out. We have redesigned caption of Figure 1 to provide a more intuitive flow of the framework.
>
> **Question 1: Is Pi-0 fine-tuned on the same dataset?**
>
> **Response:** **Yes, the Pi0 generalist policy was fine-tuned on the exact same training dataset from the RoboTwin tasks used for NoTVLA.** The key difference lies in the data processing: Pi0 uses original action trajectories, whereas NoTVLA uses our proposed kinematics-based keyframe sampling. This ensures a fair comparison of the methodology itself rather than data scale.

---

> ### Author Response · Authors · 2025-11-21
> **Response to Reviewer 41yV (Part2/2)**
>
> **Question 2: Failure modes and occlusion.**
>
> **Response:** This is a critical question for real-world deployment. We have identified two main failure modes:
>
> 1.  **Occlusion:** As you correctly surmised, if the target object or the specific anchor point is occluded from the camera's view, our method will fail to retrieve the necessary depth information.
>     *   *Mitigation:* In practice, this is often mitigated by adding a wrist-mounted camera to provide an alternative viewpoint, a standard practice in manipulation which we discussed in our response to Reviewer gus3.
> 2.  **Reflective Surfaces:** In real-robot experiments, we found that highly reflective scenes (e.g., metal surfaces) cause significant noise in depth sensors.
>     *   *Advantage of our method:* However, this actually highlights a strength of our approach. Since NoTVLA only requires **single-point depth** (the anchor) rather than a high-quality dense depth map, it is easier to filter or correct a single point's depth value than to reconstruct a full, noise-free depth map required by other 3D-centric methods. We believe this makes our approach more amenable to hardware optimizations.

---

> > ### Comment · Reviewer_41yV · 2025-11-25
> >
> > Thank you for your detailed response. As you have addressed my concerns, I have raised my score.

---

> > > ### Author Response · Authors · 2025-11-26
> > > **Re: Official Comment by Reviewer 41yV**
> > >
> > > We sincerely appreciate your positive feedback. If you have any further questions or suggestions, please do not hesitate to let us know. We greatly value your professional insights, as they are instrumental in helping us further improve the quality of our work.

---

### Official Review · Reviewer_gus3 · 2025-11-01

**Soundness:** 2
**Presentation:** 2
**Contribution:** 3
**Rating:** 6
**Confidence:** 4

**Summary:**

This paper presents an approach (NoTVLA) that trains a vision-language-action (VLA) model on an intermediary waypoint-based action space (as opposed to the raw EEF position action space). The full process of predicting actions from the VLA looks like the following: (1) given the image and language instruction, the VLA predicts the pixel XY coordinates of the next waypoint (e.g., the pixel coordinates of the object to pick up). (2) A depth sensor (e.g., an RGBD camera) is queried to obtain depth at this pixel coordinate. (3) The VLA is queried again with the image, language instruction, and now pixel XY location and depth to produce N action tokens. These tokens describe the path to take to reach from the current robot position to the waypoint predicted in the first step, with each token encoding an intermediary XY pixel location, depth prediction, gripper state, and EEF orientation. (4) Finally a motion planner is used to generate a smooth trajectory that connects these waypoints and the full trajectory segment is played open-loop. The authors argue that the reason to choose this intermediary action space instead of the original action space as most VLAs do is that (1) action prediction across robot embodiments or very diverse robot states looks more similar, which encourages feature sharing, and (2) this waypoint prediction problem is less of a departure from the original base VLM pre-training than traditional VLA training, allowing for less VLA training compute to be spent and better generalization since the model retains more of its pre-training. On a few simulated benchmarks the authors compare their approach against imitation learning (e.g., diffusion policy) and VLA (e.g., pi0) baselines and find that their approach generalizes better and gets better success rates.

**Strengths:**

(1) While the use of intermediary action spaces such as waypoints has been studied extensively and is thus not new, I like the authors’ motivation of this design decision in the context of VLAs (see summary for the motivation), though admittedly said motivations were a bit difficult to parse from the writing. In particular, the fact that VLA training tends to require a significant departure from the base VLM is not a widely discussed phenomenon (as far as I’m aware), so it is a strength that this paper has identified this problem and proposed a solution for it.

(2) The technical approach is sound and is justified by ablations (section 4.2.1), and experiments are reasonably thorough (though there seems to be no real-world experiments, which is a weakness).

(3) The empirical results agree with the motivations. Their VLA needs much less compute to train because it is less of a departure from the base VLM, and it also generalizes better than baselines like pi0 to slightly OOD task variations (table 3).

**Weaknesses:**

(1) Seemingly no real-robot experiments to validate their approach

(2) The authors claim that a strength of their approach is that their policies do not require a wrist mounted camera. However this statement can only be made if the authors have trained policies both with and without wrist mounted cameras, and empirically found that they are not needed. Also I don’t buy that wrist cameras can truly be avoided — there are many tasks for which the arms will occlude the shoulder camera, making wrist cameras necessary.

(3) In the abstract and introduction the authors frequently use the term "catastrophic forgetting”, but do not sufficiently explain in what context catastrophic forgetting is happening. Is the “forgetting” happening when the base VLM is fine-tuned into a VLA? Or is it happening when the VLA is then further fine-tuned on specific downstream tasks? My assessment after reading the paper is that the authors are referring to the former, but this could be made more clear.

(4) Several statements (concentrated especially in the introduction) are not clear, and it’s very difficult to parse what they mean. Further, many claims are simply unsubstantiated. Listing a few:

- Lines 41 to 45: “catastrophic forgetting […] is aggravated […] by an over-reliance on dense […] action trajectories […] when fine-tuned on a new task the model may overwrite previously consolidated competencies, degrading prior task performance” —> it is unclear to me what this statement means. My approximate takeaway is that the authors are trying to say that action prediction is a significant departure from the base VLM capabilities.

- Similarly lines 46-50: “The root cause of this fragility often lies in the prevailing training paradigm […]”. The authors should avoid making statements like “the root cause of X is Y” without some sort of evidence. This evidence can take the form of prior work citations, or of an easy to follow step-by-step logical argument.

- Line 53: “mitigating catastrophic forgetting” —-> this statement doesn’t explain *how* the approach mitigates catastrophic forgetting. My interpretation is that it makes the VLA action prediction task look a lot more like the base VLM pre-training tasks, but this needs to be explicitly mentioned.

- Figure 1 caption and rest of the introduction: same thing. In what context is catastrophic forgetting happening? Why do the authors think it is happening? Why does their approach solve this?

(5) The use of waypoints diminishes the applicability of the method to only pick-and-place tasks. I.e., can non-pick-and-place tasks, like T-shirt folding, stirring, or tool use be solved by the same method? How will waypoints be selected in such cases?

(6) It seems that with the proposed approach, entire trajectory segments, i.e., everything connecting two key-frames (like a gripper opening and the subsequent gripper closing), are executed completely open-loop. By open-loop I mean the VLA is not inferenced again until the motion planner executes the full trajectory segment. This is both a strength and a weakness; it is a strength in that the motion segment will likely be smooth, but it is a weakness if something unexpected happens while the motion segment is being played, like the object falling from the robot’s grasp, that would require reactivity and more closed-loop control.

(7) The acceleration threshold used to pick keyframes needs to be selected by the data-collector manually per task (line 240), which increases the burden on the data collector.

**Questions:**

(1) In Figure 4 (left), is the x-axis scale correct? The number of training steps for pi0 seems an order of magnitude too large.

(2) There are also a number of questions in the weaknesses section that I would appreciate be addressed.

---

> ### Author Response · Authors · 2025-11-21
> **Response to Reviewer gus3 (Part1/2)**
>
> We sincerely thank the reviewer for the detailed feedback, particularly regarding the definitions of catastrophic forgetting and the practicalities of real-robot deployment. We have addressed your concerns point-by-point below.
>
> **Weakness 1: Lack of real-robot experiments.**
>
> **Response:** This was an oversight in our initial presentation. We actually conducted extensive real-world experiments on both **Agilex PiPER** and **Franka** platforms. In the previous version, these results were placed in the **Appendix E**, which led to the misunderstanding. We have now added key statement to the main text to better demonstrate the effectiveness of our approach.
>
> **Table: Real-Robot Success Rates**
>
> | Robot      | Task            | SR   |
> | :--------- | :-------------- | :--- |
> | **PiPER**  | Press button    | 9/10 |
> |            | Place cube flag | 6/10 |
> |            | Stack cube two  | 2/10 |
> | **Franka** | Stack cube two  | 5/10 |
>
> **Weakness 2: Claims regarding wrist-mounted cameras.**
>
> **Response:** You raise a very valid point regarding hardware design constraints. We fully agree that in many scenarios, self-occlusion by the robot arm makes wrist cameras indispensable.
> **Clarification:** Our statement was intended to highlight the flexibility of the NoTVLA architecture—specifically, that by introducing Depth Anchors, we can complete certain tasks using only a third-person view that might traditionally require multi-view setups, thus lowering hardware barriers.
> **VLM Multi-view Challenges:** As discussed in recent works (such as *Baiqiao Yin, Qineng Wang, et al. Spatial Mental Modeling from Limited Views*), endowing VLMs with robust multi-view capabilities remains a challenge at the current stage. The complexity of fusing multiple viewpoints can sometimes hinder the model's pre-trained semantic understanding.
> **Conclusion:** We acknowledge that as VLM architectures evolve to handle multi-view inputs more effectively, using wrist-mounted cameras will undoubtedly be the better choice for resolving occlusion. Our current approach offers a pragmatic alternative that maximizes performance within the constraints of existing single-view VLM capabilities.
>
> **Weakness 3: Context of "Catastrophic Forgetting".**
>
> **Response:** You are absolutely correct, and your assessment is spot on. By "catastrophic forgetting," we are referring to the degradation of the base VLM's general capabilities when it is fine-tuned into a VLA. *VLM lost all it remembers when comes into a VLA.*
>
> **Clarification:** To clarify with a concrete example: we attempted to replicate the zero-shot generalization scenarios seen in works like RT-2, using VLA baselines like Pi0. We expected the model to perform novel tasks based on concepts it should have learned during its VLM pre-training (e.g., identifying colors, understanding spatial relationships). However, we found this to be very difficult. These models often struggled with basic concepts like color recognition that a base VLM handles easily, sometimes requiring additional fine-tuning just to re-learn them. This indicates that the fine-tuning process for dense action prediction had overwritten these fundamental capabilities. We acknowledge this point was not made clear enough in the original paper and have updated the text accordingly.
>
>
> **Weakness 4: Unclear statements and substantiation.**
>
> **Response:** Thank you for highlighting these specific lines. We have revised the introduction to be more precise:
>
> - **Clarified the cause of catastrophic forgetting:** In the first paragraph of the introduction, We have specified that when applying a pre-trained Vision-Language Model (VLM) to robot control, a significant "task conflict" arises. This conflict is between the traditional dense trajectory prediction task (a low-level, high-frequency control task) and the VLM's original pre-training objectives (high-level semantic tasks like text generation and image captioning). This inconsistency is the primary reason the model overwrites its original capabilities during fine-tuning, leading to catastrophic forgetting.
> - **Explained how NoTVLA solves this problem:** In the second paragraph and the caption for Figure 1, we have elaborated on how the NoTVLA framework reframes the action prediction task by narrowing the supervision scope to sparse, semantically-aligned keyframes. This method makes the new task more consistent with the VLM's pre-training tasks, thereby mitigating the "task conflict." It allows the model to leverage its existing reasoning abilities without overwriting them, effectively alleviating catastrophic forgetting.
> - **Adjusted the wording:** We have adopted your suggestion to avoid strong, unsubstantiated claims like "the root cause is..." and instead present the arguments through clearer logical reasoning.
>
> (see in Part2/2)

---

> ### Author Response · Authors · 2025-11-21
> **Response to Reviewer gus3 (Part2/2)**
>
> **Weakness 5: Applicability to non-pick-and-place tasks.**
>
> **Response:** The use of keyframes (waypoints) does not limit the method to pick-and-place tasks. We would like to share our perspective on why this approach generalizes:
>
> 1.  **Keyframes as Inflection Points:** Keyframes represent critical state changes or geometric constraints in a trajectory, not just grasp/release actions. For tasks like "folding a T-shirt" or "stirring," keyframes capture the critical inflection points of the motion (e.g., the start/end of a fold, or the cardinal points of a circular stirring motion). The spline interpolation then reconstructs the smooth motion between these constraints.
> 2.  **Anchors as Constraint Prompts:** We view the core challenge of VLA as establishing **a mapping from language and scene inputs to 3D manipulation trajectories**. While prior works (e.g., SoFar, Hamster) addressed start/end points or 2D trajectories, generating high-quality 3D trajectories remains difficult. In our framework, **Depth Anchors** act as a "constraint prompt." They collapse the often ill-posed 3D trajectory inference into a solution space that aligns with the real-world scene.
> 3.  **Generalization Mechanism:** For non-pick-and-place tasks, this mechanism is equally vital. For example, in a "stirring" task, the depth anchors ensure that the inferred 2D trajectory is projected onto the correct 3D plane (e.g., inside the bowl rather than floating above it). By using a few depth anchors as probes to align the VLM's inference space with the real space, we can efficiently align the critical parts of 3D trajectories for various tasks without needing dense 3D reconstruction.
>
> **Weakness 6: Open-loop execution risks.**
>
> **Response:** Your concern about open-loop execution is well-founded. However, our design does not preclude closed-loop control. Which has been shown at **Appendix D.1** in our old version.
> *   **Asynchronous Inference:** In practice, we employ an asynchronous inference strategy. While the robot executes the trajectory segment generated from the *current* keyframes, the VLM is asynchronously inferring the *next* set of keyframes based on the latest visual observation.
> *   **Reactivity:** This allows the system to adjust the plan if the object moves or slips. We have detailed this implementation in **Appendix D.1** and added a schematic to explain how this improves reactivity. This may involve tedious engineering details. If you have any doubts about this, or if our schematic diagrams are still inconvenient to understand, please feel free to let us know.
>
> **Weakness 7: Burden of manual threshold selection.**
>
> **Response:** We acknowledge this limitation. To validate this, we designed an evaluation pipeline within the RoboTwin simulation environment. After setting parameters, we can preliminarily verify the replay success rate of trajectories sampled for NoTVLA training. We found that:
>
> 1. **High Success Rate for Pick-and-Place:** Tasks based on gripper state changes already achieve very high success rates.
> 2. **Optimization for Complex Tasks:** For non-Pick-and-Place tasks, manual parameter adjustment can boost trajectory replay success rates to over 80%.
> 3. **Annotation Tools:** We have also developed visual data annotation tools to further reduce the workload of trajectory labeling.
>
> Looking ahead, we believe that incorporating **tactile information** will be a crucial direction for future robotics research. We are eager to further optimize our dataset processing pipeline based on multi-modal perception fusion (including tactile data) to reduce reliance on manual thresholds and improve scalability.
>
> **Question 1: X-axis scale in Figure 4 (left).**
>
> **Response:** Thank you for your keen observation. This was indeed a plotting error. In the original figure, we summed the total training steps of the generalist model (Pi0) across all tasks and compared it directly to single-task steps, resulting in a misleading order-of-magnitude difference. We have corrected the figure to use "average training steps per task" or normalized resources for a fair comparison. The updated figure is now in the main text.
>
> **Question 2: Addressing other weaknesses.**
>
> **Response:** We have addressed all points raised in the "Weaknesses" section above. We hope these responses clarify your doubts.

---

> > ### Comment · Reviewer_gus3 · 2025-11-24
> >
> > I appreciate the authors' efforts in updating the paper towards addressing my concerns.
> >
> > Among the items I had originally listed under weaknesses, (3), (4), and (6) have been resolved by the authors' rebuttal, and the authors gave satisfying answers for (2) and (7). The explanation of the method is indeed much more clear in the current paper draft -- nice job!
> >
> > Regarding weakness (1), real-world experiments, while I see the table in the appendix, as far as I was able to tell this real-world results table does not appear in the main body (which it should). Also, the table doesn't compare against any baselines -- it only shows the performance of the proposed method.
> >
> > And regarding weakness (5), while the argument the authors gave in the rebuttal may make sense, ultimately robot experiments on those non-pick-and-place tasks like t-shirt folding/stirring would be needed to validate the claim that the approach can be applied even in more dextrous manipulation settings.
> >
> > Taking everything into account, in my opinion the quality of the paper has improved, and so I will raise my score to 8.

---

> > > ### Author Response · Authors · 2025-11-26
> > > **Re: Official Comment by Reviewer gus3**
> > >
> > > We sincerely appreciate your professional evaluation and constructive feedback.
> > >
> > > We fully agree that rigorous real-world validation is crucial. We will move the real-world experimental results to the main body of the paper. Furthermore, we will expand these experiments to include comparisons against both classic VLA baselines and specialized expert models to provide a more comprehensive assessment.
> > >
> > > Regarding more complex, dexterous tasks (such as folding or stirring), we agree that experimental validation is needed. We identify **keyframe annotation** as the primary challenge in scaling to these domains. We are actively initiating engineering efforts to explore tactile-based and learning-based annotation paradigms to increase automation, which may enable us to fully tackle these scenarios in future work.
> > >
> > > Thank you again for your recognition and your professional insight in robotics. We remain open to any further questions and look forward to continuing to improve our work based on your expert guidance.

---

### Official Review · Reviewer_FiUa · 2025-11-04

**Soundness:** 4
**Presentation:** 3
**Contribution:** 3
**Rating:** 6
**Confidence:** 3

**Summary:**

This paper proposed Narrowing of Trajectory VLA (NoTVLA), which tries to solve the catastrophic forgetting issues of fine-tuning VLA models. The paper attributes this issue to the dense trajectory representation, and addresses the issue by applying Keyframe Selection on training data, and performing Anchor Point prediction (APP) with Anchor-conditioned token generation (ACTG) during inference. The actions are decoded with spline-based detokenizer for smooth and high-frequency trajectories. Experiments demonstrate its effectiveness.

**Strengths:**

1. The paper is generally well-written and clear.
2. The paper provides very extensive experiments in both simulation and reality, that show promising results on NoTVLA’s manipulation capabilities, generalizability, and the preserved language reasoning capabilities. The system is also training-efficient.3
3. The analysis provides rich insights on the language understanding degradation, zero-shot generalization, training efficiency, *e.t.c*.

**Weaknesses:**

##

1. The Kinematics-Based Keyframe Selection relies on an acceleration threshold $\alpha$ that is "hand-designed per task." This introduces a manual, expert-in-the-loop step that could hinder scalability.
2. The APP and ACTG modules requires the depth information of the scene. While the VLA system has a good performance, a sensitivity analysis on the depth noise or bias is necessary to show how the system would work with different depth sensors and environments.
3. The 2D anchor representation could limit some tasks. Additional 2D or 3D representation (*e.g.*, via visual visual prompting, robot-object constraints, *e.t.c.*) could be discussed for comprehensiveness.
4. Some editorial and writing issues
    1. While the experiments validate the hypothesis of “dense fine-tuning is the cause of catastrophic forgetting”, the reason or the mechanism is not intuitively explained, especially at the beginning of the paper. This causes confusion and doubts.
    2. Figure and table captions are not self-contained.

**Questions:**

The language capabilities degradation, while remedies, still occurs as suggests by Tab. 6. Do you have any insight into which specific language reasoning skills (e.g., spatial, logical, mathematical) are most affected by this sparse fine-tuning?

---

> ### Author Response · Authors · 2025-11-21
> **Response to Reviewer FiUa (Part1/2)**
>
> We sincerely thank the reviewer for the insightful comments and constructive feedback. We have carefully addressed each point below.
>
> **Weakness 1: Reliance on hand-designed acceleration threshold $\alpha$ and scalability concerns.**
> **Response:** Thank you for raising this point. While simple tasks often have automated scripts available, we agree that scalability for complex tasks is a challenge. In our current work, we have implemented some practical engineering optimizations. For simple Pick-and-Place tasks, gripper state changes are often sufficient for keyframe identification. For non-Pick-and-Place tasks, we utilize the acceleration threshold.
>
> To validate this, we designed an evaluation pipeline within the RoboTwin simulation environment. After setting parameters, we can preliminarily verify the replay success rate of trajectories sampled for NoTVLA training. We found that:
>
> 1. **High Success Rate for Pick-and-Place:** Tasks based on gripper state changes already achieve very high success rates.
> 2. **Optimization for Complex Tasks:** For non-Pick-and-Place tasks, manual parameter adjustment can boost trajectory replay success rates to over 80%.
> 3. **Annotation Tools:** We have also developed visual data annotation tools to further reduce the workload of trajectory labeling.
>
> Looking ahead, we believe that incorporating **tactile information** will be a crucial direction for future robotics research. We are eager to further optimize our dataset processing pipeline based on multi-modal perception fusion (including tactile data) to reduce reliance on manual thresholds and improve scalability.
>
> **Weakness 2: Sensitivity analysis on depth noise or bias for APP and ACTG modules.**
>
> **Response:** This is a critical question. To address this, we have conducted additional experiments to analyze the system's sensitivity to depth noise.
>
> **Table: Success Rate under Different Depth Noise Levels**
>
> | Task / Noise Level (cm) | 0.00 | 1.00 | 10.00 | 100.00 |
> |---|---:|---:|---:|---:|
> | place_a2b_left | 0.40 | 0.35 | 0.20 | 0.10 |
> | place_a2b_right | 0.30 | 0.25 | 0.15 | 0.05 |
> | place_object_scale | 0.45 | 0.25 | 0.25 | 0.15 |
> | stack_blocks_three | 0.40 | 0.30 | 0.00 | 0.00 |
> | stack_bowl_three | 0.80 | 0.40 | 0.10 | 0.10 |
> | blocks_ranking_size | 0.55 | 0.20 | 0.00 | 0.00 |
> | blocks_ranking_rgb | 0.65 | 0.35 | 0.05 | 0.10 |
>
> *Note: Noise level represents the mean absolute value of depth noise. The noise follows a normal distribution with random positive/negative bias. Current state-of-the-art depth estimation models typically have errors in the range of 1cm–10cm for tabletop scenarios.*
>
> **Observations:**
>
> 1. **Robustness at Low Noise:** When the maximum noise is around 1 cm, the task performance remains comparable to the noise-free baseline.
> 2. **Degradation at Medium Noise:** At 10 cm noise, most tasks show a noticeable decline in success rate.
> 3. **Failure at High Noise:** At 1 m noise, the system fails almost completely.
>
> **Discussion:** These results suggest that while noise inevitably impacts performance, our model maintains robustness within the error range of typical depth sensors (1-10cm). Interestingly, in some tabletop scenarios similar to the training distribution, the model can still complete tasks even with significant noise, indicating it has learned some implicit depth estimation capabilities. However, for complex scenes, precise depth input remains critical due to the ill-posed nature of depth estimation.
>
> Unlike traditional 3D-based models (e.g., DP3) that require dense point clouds, our method only requires one or two depth anchors as spatial probes to align with the VLM's coarse spatial planning. Current single-point laser ranging devices (e.g., SK60) can achieve extremely high precision ($\pm 2$mm) in tabletop settings, making our approach highly feasible for hardware optimization.
>
> **Weakness 3: Limitations of 2D anchor representation and discussion on additional representations.**
>
> **Response:** This is an excellent point, also discussed in related works like SoFar and Hamster. We would like to share our perspective:
>
> The core challenge of VLA is mapping language and scene inputs to 3D trajectories. While prior works solved start/end points (SoFar) or 2D trajectories (Hamster), generating high-quality 3D trajectories remains difficult.
>
> 1. **Our Approach:** We rely on depth anchors to align the terminal part of the 3D trajectory. This acts as a constraint prompt, collapsing the ill-posed 3D inference into a solution space that fits the real world. This allows for easy deployment in simulation and real-world settings without rigid engineering (e.g., specific drawer handling in SoFar).
> 2. **Additional Representations:** We agree that additional 2D or 3D representations (visual prompts, constraints) serve a similar purpose—providing visual-semantic constraints. However, one must balance precision and convenience.
>
> (next in Part2/2)

---

> ### Author Response · Authors · 2025-11-21
> **Response to Reviewer FiUa (Part2/2)**
>
> 3. **Why Sparse Works:** We believe that for simple tasks like grasping, kinematics-based keyframes retain the most critical information for success. The dense trajectory segments between keyframes have less impact on success but burden the model with high-frequency control signals that conflict with its pre-trained language capabilities. By learning sparsely away from keyframes and densely near them (to ensure precision, e.g., collision avoidance), NoTVLA achieves good task performance while protecting the model's general capabilities.
>
> **Question: Insight into specific language reasoning skills affected by sparse fine-tuning.**
>
> **Response:** This is a very insightful question. While NoTVLA significantly mitigates catastrophic forgetting, a slight degradation persists. Based on our detailed evaluation on C-Eval, CMMLU, and MMLU (as shown in the table above), we offer the following insights:
>
> 1. **Abstract Logic and Mathematics are Most Affected:** We observed that tasks involving pure abstract symbolic reasoning (e.g., mathematical problem solving, complex logical deduction) tend to be the most affected areas. We believe this is because the distribution of manipulation training data (physical action sequences) diverges most significantly from the semantic space of abstract logic. When the model is forced to map visual inputs to specific physical actions, the high-dimensional feature space originally used for abstract processing is compressed.
> 2. **Nuanced Impact on Spatial Reasoning:** Interestingly, spatial reasoning capabilities appear to be less affected. We hypothesize that this is because our training data inherently contains rich 3D spatial information (via anchors and trajectories), which aligns with, rather than conflicts with, the VLM's pre-existing spatial understanding.
> 3. **Protective Effect of Sparse Fine-tuning:** Despite the slight decline, the contrast is striking. When trained with Dense action, logic scores plummeted by ~10-15 points. With NoTVLA (5 points), scores dropped not so much. This demonstrates that our sparse keyframe strategy successfully decouples "manipulation learning" from "general reasoning" in the latent space, maximally preserving core cognitive skills.

---

### Note · Program_Chairs · 2026-01-17
**Submission Desk Rejected by Program Chairs**

The following references in this submission do not refer to real documents and/or have major errors in bibliographic information:

 Zihan Zhao, Kuan Zhang, Tete Chen, and Pieter Abbeel. Learning action chunking with transformers for long-horizon task completion. In Conference on Robot Learning (CoRL), pp. 1706-1716. PMLR, 2022.
F. Liu, Y. Zhang, et al. Llm+robot: Large language models for robotics planning and skill learning. arXiv preprint arXiv:2304.12244, 2023b.
A. Intelligence. A dual-system framework for general-purpose agents, 2025.